# Contextual Stochastic Block Models

**Yash Deshpande**[*]    Andrea Montanari [†]    Elchanan Mossel[‡]    Subhabrata Sen[§]

## Abstract

We provide the first information theoretic tight analysis for inference of latent community structure given a sparse graph along with high dimensional node covariates, correlated with the same latent communities. Our work bridges recent theoretical breakthroughs in the detection of latent community structure without nodes covariates and a large body of empirical work using diverse heuristics for combining node covariates with graphs for inference. The tightness of our analysis implies in particular, the information theoretical necessity of combining the different sources of information. Our analysis holds for networks of large degrees as well as for a Gaussian version of the model.

## 1   Introduction

Data clustering is a widely used primitive in exploratory data analysis and summarization. These methods discover clusters or partitions that are assumed to reflect a latent partitioning of the data with semantic significance. In a machine learning pipeline, results of such a clustering may then be used for downstream supervised tasks, such as feature engineering, privacy-preserving classification or fair allocation [CMS11, KGB+12, CDPF+17].

At risk of over-simplification, there are two settings that are popular in literature. In *graph clustering*, the dataset of $n$ objects is represented as a symmetric *similarity matrix* $A = (A_{ij})_{1 \leq i,j \leq n}$. For instance, $A$ can be binary, where $A_{ij} = 1$ (or 0) denotes that the two objects $i$, $j$ are similar (or not). It is, then, natural to interpret $A$ as the adjacency matrix of a graph. This can be carried over to non-binary settings by considering weighted graphs. On the other hand, in more traditional (binary) *classification* problems, the $n$ objects are represented as $p$-dimensional feature or covariate vectors $b_1, b_2, \cdots, b_n$. This feature representation can be the input for a clustering method such as $k$-means, or instead used to construct a similarity matrix $A$, which in turn is used for clustering or partitioning. These two representations are often taken to be mutually exclusive and, in fact, interchangeable. Indeed, just as feature representations can be used to construct similarity matrices, popular spectral methods [NJW02, VL07] implicitly construct a low-dimensional feature representation from the similarity matrices.

This paper is motivated by scenarios where the graph, or similarity, representation $A \in \mathbb{R}^{n \times n}$, and the feature representation $B = [b_1, b_2, \ldots, b_n] \in \mathbb{R}^{p \times n}$ provide *independent*, or *complementary*, information on the latent clustering of the $n$ objects. (Technically, we will assume that $A$ and $B$ are conditionally independent given the node labels.) We argue that in fact in almost all practical graph clustering problems, feature representations provide complementary information of the latent clustering. This is indeed the case in many social and biological networks, see e.g. [NC16] and references within.

As an example, consider the 'political blogs' dataset [AG05]. This is a directed network of political blogs during the 2004 US presidential election, with a link between two blogs if one referred to the

---

[*]Department of Mathematics, Massachusetts Institute of Technology
[†]Departments of Electrical Engineering and Statistics, Stanford University
[‡]Department of Mathematics, Massachusetts Institute of Technology
[§]Department of Mathematics, Massachusetts Institute of Technology

other. It is possible to just use the graph structure in order to identify political communities (as was done in [AG05]). Note however that much more data is available. For example we may consider an alternative feature representation of the blogs, wherein each blog is converted to a 'bag-of words' vector of its content. This gives a quite different, and complementary representation of blogs that plausibly reflects their political leaning. A number of approaches can be used for the simple task of predicting leaning from the graph information (or feature information) individually. However, given access to both sources, it is challenging to combine them in a principled fashion.

In this context, we introduce a simple statistical model of complementary graph and high-dimensional covariate data that share latent cluster structure. This model is an intuitive combination of two well-studied models in machine learning and statistics: the *stochastic block model* and the *spiked covariance model* [Abb17, HLL83, JL04]. We focus on the task of uncovering this latent structure and make the following contributions:

**Sharp thresholds:** We establish a sharp information-theoretic threshold for detecting the latent structure in this model. This threshold is based on non-rigorous, but powerful, techniques from statistical physics.

**Rigorous validation:** We consider a certain 'Gaussian' limit of the statistical model, which is of independent interest. In this limit, we rigorously establish the correct information-theoretic threshold using novel Gaussian comparison inequalities. We further show convergence to the Gaussian limit predictions as the density of the graph diverges.

**Algorithm:** We provide a simple, iterative algorithm for inference based on the belief propagation heuristic. For data generated from the model, we empirically demonstrate that the the algorithm achieves the conjectured information-theoretic threshold.

The rest of the paper is organized as follows. The model and results are presented in Section 2. Further related work is discussed in Section 3. The prediction of the threshold from statistical physics techniques is presented in 4, along with the algorithm. While all proofs are presented in the appendix, we provide an overview of the proofs of our rigorous results in Section 5. Finally, we numerically validate the prediction in Section 6.

## 2 Model and main results

We will focus on the simple case where the $n$ objects form two latent clusters of approximately equal size, labeled $+$ and $-$. Let $v \in \{\pm 1\}^n$ be the vector encoding this partitioning. Then, the observed data is a pair of matrices $(A^G, B)$, where $A^G$ is the adjacency matrix of the graph $G$ and $B \in \mathbb{R}^{p \times n}$ is the matrix of covariate information. Each column $b_i$, $i \leq n$ of matrix $B$ contains the covariate information about vertex $i$. We use the following probabilistic model: conditional on $v$, and a latent vector $u \sim \mathsf{N}(0, I_p/p)$:

$$\mathbb{P}(A_{ij}^G = 1) = \begin{cases} c_{\text{in}}/n & \text{with probability }, \\ c_{\text{out}}/n & \text{otherwise.} \end{cases} \tag{1}$$

$$b_i = \sqrt{\frac{\mu}{n}} v_i u + \frac{Z_i}{\sqrt{p}}, \tag{2}$$

where $Z_i \in \mathbb{R}^p$ has independent standard normal entries. It is convenient to parametrize the edge probabilities by the average degree $d$ and the normalized degree separation $\lambda$:

$$c_{\text{in}} = d + \lambda\sqrt{d}, \qquad c_{\text{out}} = d - \lambda\sqrt{d}. \tag{3}$$

Here $d$, $\lambda$, $\mu$ are parameters of the model which, for the sake of simplicity, we assume to be fixed and known. In other words, two objects $i, j$ in the same cluster or community are *slightly more likely* to be connected than for objects $i, j'$ in different clusters. Similarly, according to (2), they have *slightly positively* correlated feature vectors $b_i$, $b_j$, while objects $i, j'$ in different clusters have negatively correlated covariates $b_i, b_{j'}$.

Note that this model is a combination of two observation models that have been extensively studied: the stochastic block model and the spiked covariance model. The stochastic block model has its roots in sociology literature [HLL83] and has witnessed a resurgence of interest from the computer

science and statistics community since the work of Decelle et al. [DKMZ11]. This work focused on the sparse setting where the graph as $O(n)$ edges and conjectured, using the non-rigorous cavity method, the following *phase transition* phenomenon. This was later established rigorously in a series of papers [MNS15, MNS13, Mas14].

**Theorem 1** ([MNS15, MNS13, Mas14]). *Suppose $d > 1$ is fixed. The graph $G$ is distinguishable with high probability from an Erdös-Renyi random graph with average degree $d$ if and only if $\lambda \geq 1$. Moreover, if $\lambda > 1$, there exists a polynomial-time computable estimate $\widehat{v} = \widehat{v}(A^G) \in \{\pm 1\}^n$ of the cluster assignment satisfying, almost surely:*

$$\liminf_{n \to \infty} \frac{|\langle \widehat{v}, v \rangle|}{n} \geq \varepsilon(\lambda) > 0. \tag{4}$$

In other words, given the graph $G$, it is possible to non-trivially estimate the latent clustering $v$ if, and only if, $\lambda > 1$.

The covariate model (2) was proposed by Johnstone and Lu [JL04] and has been extensively studied in statistics and random matrix theory. The weak recovery threshold was characterized by a number of authors, including Baik et al [BBAP05], Paul [Pau07] and Onatski et al [OMH+13].

**Theorem 2** ([BBAP05, Pau07, OMH+13]). *Let $\widehat{v}_1$ be the principal eigenvector of $B^\mathsf{T} B$, where $\widehat{v}_1$ is normalized so that $\|\widehat{v}_1\|^2 = n$. Suppose that $p, n \to \infty$ with $p/n \to 1/\gamma \in (0, \infty)$. Then $\liminf_{n \to \infty} |\langle \widehat{v}_1, v \rangle|/n > 0$ if and only if $\mu > \sqrt{\gamma}$. Moreover, if $\mu < \sqrt{\gamma}$, no such estimator exists.*

In other words, this theorem shows that it is possible to estimate $v$ solely from the covariates using, in fact, a spectral method if, and only if $\mu > \sqrt{\gamma}$.

Our first result is the following prediction that establishes the analogous threshold prediction that smoothly interpolates between Theorems 1 and 2.

**Claim 3** (Cavity prediction). *Given $A^G, B$ as in Eqs.(1), (2), and assume that $n, p \to \infty$ with $p/n \to 1/\gamma \in (0, \infty)$. Then there exists an estimator $\widehat{v} = \widehat{v}(A^G, B) \in \{\pm 1\}^n$ so that $\liminf |\langle \widehat{v}, v \rangle|/n$ is bounded away from 0 if and only if*

$$\lambda^2 + \frac{\mu^2}{\gamma} > 1. \tag{5}$$

We emphasize here that this claim is *not rigorous*; we obtain this prediction via the cavity method. The cavity method is a powerful technique from the statistical physics of mean field models [MM09]. Our instantiation of the cavity method is outlined in Section 4, along with Appendix B and D (see supplement). The cavity method is remarkably successful and a number of its predictions have been made rigorous [MM09, Tal10]. Consequently, we view Claim 3 as a conjecture, with strong positive evidence. Theorems 1 and 2 confirm the cavity prediction rigorously in the corner cases, in which either $\lambda$ or $\mu$ vanishes, using intricate tools from random matrix theory and sparse random graphs.

Our main result confirms rigorously Claim 3 in the limit of large degrees.

**Theorem 4.** *Suppose $v$ is uniformly distributed in $\{\pm 1\}^n$ and we observe $A^G, B$ as in (1), (2). Consider the limit $p, n \to \infty$ with $p/n \to 1/\gamma$. Then, for some $\varepsilon(\lambda, \mu) > 0$ independent of $d$,*

$$\liminf_{n \to \infty} \sup_{\widehat{v}(\cdot)} \frac{|\langle \widehat{v}(A^G, B), v \rangle|}{n} \geq \varepsilon(\lambda, \mu) - o_d(1) \qquad \text{if } \lambda^2 + \mu^2/\gamma > 1, \tag{6}$$

$$\limsup_{n \to \infty} \sup_{\widehat{v}(\cdot)} \frac{|\langle \widehat{v}(A^G, B), v \rangle|}{n} = o_d(1) \qquad \text{if } \lambda^2 + \mu^2/\gamma < 1. \tag{7}$$

*Here the limits hold in probability, the supremum is over estimators $\widehat{v} : (A^G, B) \mapsto \widehat{v}(A^G, B) \in \mathbb{R}^n$, with $\|\widehat{v}(A^G, B)\|_2 = \sqrt{n}$. Here $o_d(1)$ indicates a term independent of $n$ which tends to zero as $d \to \infty$.*

In order to establish this result, we consider a modification of the original model in (1), (2), which is of independent interest. Suppose, conditional on $v \in \{\pm 1\}$ and the latent vector $u$ we observe $(A, B)$ as follows:

$$A_{ij} \sim \begin{cases} \mathsf{N}(\lambda v_i v_j/n, 1/n) & \text{if } i < j \\ \mathsf{N}(\lambda v_i v_j/n, 2/n) & \text{if } i = j, \end{cases} \tag{8}$$

$$B_{ai} \sim \mathsf{N}(\sqrt{\mu} v_i u_a/\sqrt{n}, 1/p). \tag{9}$$

This model differs from (1), in that the graph observation $A^G$ is replaced by the observation $A$ which is equal to $\lambda vv^{\mathsf{T}}/n$, corrupted by Gaussian noise. This model generalizes so called 'rank-one deformations' of random matrices [Péc06, KY13, BGN11], as well as the $\mathbb{Z}_2$ synchronization model [ABBS14, Cuc15].

Our main motivation for introducing the Gaussian observation model is that it captures the large-degree behavior of the original graph model. The next result formalizes this intuition: its proof is an immediate generalization of the Lindeberg interpolation method of [DAM16].

**Theorem 5.** *Suppose $v \in \{\pm1\}^n$ is uniformly random, and $u$ is independent. We denote by $I(v; A^G, B)$ the* mutual information *of the latent random variables $v$ and the observable data $A^G, B$. For all $\lambda, \mu$: we have that:*

$$\lim_{d\to\infty} \limsup_{n\to\infty} \frac{1}{n}|I(v; A^G, B) - I(v; A, B)| = 0, \tag{10}$$

$$\lim_{d\to\infty} \limsup_{n\to\infty} \left|\frac{1}{n}\frac{\mathrm{d}I(v; A^G, B)}{\mathrm{d}(\lambda^2)} - \frac{1}{4}\mathsf{MMSE}(v; A^G, B)\right| = 0, \tag{11}$$

*where* $\mathsf{MMSE}(v; A^G, B) = n^{-2}\mathbb{E}\{\|vv^{\mathsf{T}} - \mathbb{E}\{vv^{\mathsf{T}}|A^G, B\}\|_F^2\}$.

For the Gaussian observation model (8), (9) we can establish a precise weak recovery threshold, which is the main technical novelty of this paper.

**Theorem 6.** *Suppose $v$ is uniformly distributed in $\{\pm1\}^n$ and we observe $A, B$ as in (8), (9). Consider the limit $p, n \to \infty$ with $p/n \to 1/\gamma$.*

1. *If $\lambda^2 + \mu^2/\gamma < 1$, then for any estimator $\widehat{v} : (A, B) \mapsto \widehat{v}(A, B)$, with $\|\widehat{v}(A, B)\|_2 = \sqrt{n}$, we have $\limsup_{n\to\infty} |\langle\widehat{v}, v\rangle|/n = 0$.*

2. *If $\lambda^2 + \mu^2/\gamma > 1$, let $\widehat{v}(A, B)$ be normalized so that $\|\widehat{v}(A, B)\|_2 = \sqrt{n}$, and proportional the maximum eigenvector of the matrix $M(\xi_*)$, where*

$$M(\xi) = A + \frac{2\mu^2}{\lambda^2\gamma^2\xi} B^{\mathsf{T}}B + \frac{\xi}{2} I_n, \tag{12}$$

*and $\xi_* = \arg\min_{\xi>0} \lambda_{\max}(M(\xi))$. Then, $\liminf_{n\to\infty} |\langle\widehat{v}, v\rangle|/n > 0$ in probability.*

Theorem 4 is proved by using this threshold result, in conjunction with the universality Theorem 5.

## 3    Related work

The need to incorporate node information in graph clustering has been long recognized. To address the problem, diverse clustering methods have been introduced— e.g. those based on generative models [NC16, Hof03, ZVA10, YJCZ09, KL12, LM12, XKW+12, HL14, YML13], heuristic model free approaches [BVR17, ZLZ+16, GVB12, ZCY09, NAJ03, GFRS13, DV12, CZY11, SMJZ12, SZLP16], Bayesian methods [CB10, BC11] etc. [BCMM15] surveys other clustering methods for graphs with node and edge attributes. Semisupervised graph clustering [Pee12, EM12, ZMZ14], where labels are available for a few vertices are also somewhat related to our line of enquiry. The literature in this domain is quite vast and extremely diffuse, and thus we do not attempt to provide an exhaustive survey of all related attempts in this direction.

In terms of rigorous results, [AJC14, LMX15] introduced and analyzed a model with informative edges, but they make the strong and unrealistic requirement that the label of individual edges and each of their endpoints are uncorrelated and are only able to prove one side of their conjectured threshold. The papers [BVR17, ZLZ+16] –among others– rigorously analyze specific heuristics for clustering and provide some guarantees that ensure consistency. However, these results are not optimal. Moreover, it is possible that they only hold in the regime where using either the node covariates or the graph suffices for inference.

Several theoretical works [KMS16, MX16] analyze the performance of local algorithms in the semi-supervised setting, i.e., where the true labels are given for a small fraction of nodes. In particular [KMS16] establishes that for the two community sparse stochastic block model, correlated recovery is impossible given any vanishing proportion of nodes. Note that this is in stark contrast to Theorem 4

(and the Claim for the sparse graph model) above, which posits that given high dimensional covariate information actually shifts the information theoretic threshold for detection and weak recovery. The analysis in [KMS16, MX16] is also local in nature, while our algorithms and their analysis go well beyond the diameter of the graph.

## 4 Belief propagation: algorithm and cavity prediction

Recall the model (1), (2), where we are given the data $(A^G, B)$ and our task is to infer the latent community labels $v$. From a Bayesian perspective, a principled approach computes posterior expectation with respect to the conditional distribution $\mathbb{P}(v, u|A^G, B) = \mathbb{P}(v, u, A^G, B)/\mathbb{P}(A^G, B)$. This is, however, not computationally tractable because it requires to marginalize over $v \in \{+1, -1\}^n$ and $u \in \mathbb{R}^p$. At this point, it becomes necessary to choose an approximate inference procedure, such as variational inference or mean field approximations [WJ+08]. In Bayes inference problem on locally-tree like graphs, belief propagation is optimal among local algorithms (see for instance [DM15] for an explanation of why this is the case).

The algorithm proceeds by computing, in an iterative fashion *vertex messages* $\eta_i^t, m_a^t$ for $i \in [n]$, $a \in [p]$ and *edge messages* $\eta_{i \to j}^t$ for all pairs $(i, j)$ that are connected in the graph $G$. For a vertex $i$ of $G$, we denote its neighborhood in $G$ by $\partial i$. Starting from an initialization $(\eta^{t_0}, m^{t_0})_{t_0=-1,0}$, we update the messages in the following *linear* fashion:

$$\eta_{i \to j}^{t+1} = \sqrt{\frac{\mu}{\gamma}}(B^\mathsf{T} m^t)_i - \frac{\mu}{\gamma}\eta_i^{t-1} + \frac{\lambda}{\sqrt{d}}\sum_{k \in \partial i \setminus j} \eta_{k \to i}^t - \frac{\lambda\sqrt{d}}{n}\sum_{k \in [n]} \eta_k^t, \tag{13}$$

$$\eta_i^{t+1} = \sqrt{\frac{\mu}{\gamma}}(B^\mathsf{T} m^t)_i - \frac{\mu}{\gamma}\eta_i^{t-1} + \frac{\lambda}{\sqrt{d}}\sum_{k \in \partial i} \eta_{k \to i}^t - \frac{\lambda\sqrt{d}}{n}\sum_{k \in [n]} \eta_k^t, \tag{14}$$

$$m^{t+1} = \sqrt{\frac{\mu}{\gamma}}B\eta^t - \mu m^{t-1}. \tag{15}$$

Here, and below, we will use $\eta^t = (\eta_i^t)_{i \in [n]}$, $m^t = (m_a^t)_{a \in [p]}$ to denote the vectors of vertex messages. After running the algorithm for some number of iterations $t_{\max}$, we return, as an estimate, the sign of the vertex messages $\eta_i^{t_{\max}}$, i.e.

$$\widehat{v}_i(A^G, B) = \mathrm{sgn}(\eta_i^{t_{\max}}). \tag{16}$$

These update equations have a number of intuitive features. First, in the case that $\mu = 0$, i.e. we have no covariate information, the edge messages become:

$$\eta_{i \to j}^{t+1} = \frac{\lambda}{\sqrt{d}}\sum_{k \in \partial i \setminus j} \eta_{k \to i}^t - \frac{\lambda\sqrt{d}}{n}\sum_{k \in [n]} \eta_k^t, \tag{17}$$

which corresponds closely to the spectral power method on the *nonbacktracking walk* matrix of $G$ [KMM+13]. Conversely, when $\lambda = 0$, the updates equations on $m^t, \eta^t$ correspond closely to the usual power iteration to compute singular vectors of $B$.

We obtain this algorithm from belief propagation using two approximations. First, we linearize the belief propagation update equations around a certain 'zero information' fixed point. Second, we use an 'approximate message passing' version of the belief propagation updates which results in the addition of the memory terms in Eqs. (13), (14), (15). The details of these approximations are quite standard and deferred to Appendix D. For a heuristic discussion, we refer the interested reader to the tutorials [Mon12, TKGM14] (for the Gaussian approximation) and the papers [DKMZ11, KMM+13] (for the linearization procedure).

As with belief propagation, the behavior of this iterative algorithm, in the limit $p, n \to \infty$ can be tracked using a distributional recursion called *density evolution*.

**Definition 1** (Density evolution). *Let $(\bar{m}, U)$ and $(\bar{\eta}, V)$ be independent random vectors such that $U \sim \mathsf{N}(0, 1)$, $V \sim \mathrm{Uniform}(\{\pm 1\})$, $\bar{m}, \bar{\eta}$ have finite variance. Further assume that $(\bar{\eta}, V) \stackrel{\mathrm{d}}{=} (-\bar{\eta}, -V)$ and $(\bar{m}, U) \stackrel{\mathrm{d}}{=} (-\bar{m}, -U)$ (where $\stackrel{\mathrm{d}}{=}$ denotes equality in distribution).*

We then define new random pairs $(\bar{m}', U')$ and $(\bar{\eta}', V')$, where $U' \sim \mathsf{N}(0,1)$, $V' \sim \mathrm{Uniform}(\{\pm 1\})$, and $(\bar{\eta}, V) \stackrel{\mathrm{d}}{=} (-\bar{\eta}, -V)$, $(\bar{m}, U) \stackrel{\mathrm{d}}{=} (-\bar{m}, -U)$, via the following distributional equation

$$\bar{m}'\big|_{U'} \stackrel{\mathrm{d}}{=} \mu \mathbb{E}\{V\bar{\eta}\}U' + \left(\mu \mathbb{E}\{\bar{\eta}^2\}\right)^{1/2}\zeta_1, \tag{18}$$

$$\bar{\eta}'\big|_{V'=+1} \stackrel{\mathrm{d}}{=} \frac{\lambda}{\sqrt{d}}\Big[\sum_{k=1}^{k_+} \bar{\eta}_k\big|_+ + \sum_{k=1}^{k_-} \bar{\eta}_k\big|_-\Big] - \lambda\sqrt{d}\mathbb{E}\{\bar{\eta}\}$$

$$+ \frac{\mu}{\gamma}\mathbb{E}\{U\bar{m}\} + \left(\frac{\mu}{\gamma}\mathbb{E}\{\bar{m}^2\}\right)^{1/2}\zeta_2. \tag{19}$$

Here we use the notation $X|_Y \stackrel{\mathrm{d}}{=} Z$ to mean that the conditional distribution of $X$ given $Y$ is the same as the (unconditional) distribution of $Z$. Notice that the distribution of $\bar{\eta}'\big|_{V'=-}$ is determined by the last equation using the symmetry property. Further $\bar{\eta}_k|_+$ and $\bar{\eta}_k|_-$ denote independent random variables distributed (respectively) as $\bar{\eta}|_{V=+}$ and $\bar{\eta}|_{V=-}$. Finally $k_+ \sim \mathsf{Poiss}(d/2 + \lambda\sqrt{d}/2)$, $k_- \sim \mathsf{Poiss}(d/2 - \lambda\sqrt{d}/2)$, $\zeta_1 \sim \mathsf{N}(0,1)$ and $\zeta_2 \sim \mathsf{N}(0,1)$ are mutually independent and independent from the previous random variables.

The density evolution map, denoted by $\mathsf{DE}$, is defined as the mapping from the law of $(\bar{\eta}, V, \bar{m}, U)$ to the law of $(\bar{\eta}', V', \bar{m}', U')$. With a slight abuse of notation, we will omit $V, U$, $V', U'$, whose distribution is left unchanged and write

$$(\bar{\eta}', \bar{m}') = \mathsf{DE}(\bar{\eta}, \bar{m}). \tag{20}$$

The following claim is the core of the cavity prediction. It states that the density evolution recursion faithfully describes the distribution of the iterates $\eta^t, m^t$.

**Claim 7.** *Let* $(\bar{\eta}^0, V)$, $(\bar{m}^0, U)$ *be random vectors satisfying the conditions of definition 1. Define the density evolution sequence* $(\bar{\eta}^t, \bar{m}^t) = \mathsf{DE}^t(\bar{\eta}^0, \bar{m}^0)$, *i.e. the result of iteratively applying the mapping* $\mathsf{DE}$ $t$ *times.*

*Consider the linear message passing algorithm of Eqs. (13) to (15), with the following initialization. We set* $(m_r^0)_{r \in [p]}$ *conditionally independent given* $u$, *with conditional distribution* $m_r^0|_u \stackrel{\mathrm{d}}{=} \bar{m}^0|_{U=\sqrt{p}u_r}$. *Analogously,* $\eta_i^0, \eta_{i \to j}^0$ *are conditionally independent given* $v$ *with* $\eta_i^0|_v \stackrel{\mathrm{d}}{=} \bar{\eta}^0|_{V=v_i}$, $\eta_{i \to j}^0|_v \stackrel{\mathrm{d}}{=} \bar{\eta}^0|_{V=v_i}$. *Finally* $\eta_i^{-1} = \eta_{i \to j}^{-1} = m_r^{-1} = 0$ *for all* $i, j, r$.

*Then, as* $n, p \to \infty$ *with* $p/n \to 1/\gamma$, *the following holds for uniformly random indices* $i \in [n]$ *and* $a \in [p]$:

$$(m_a^t, u_a\sqrt{p}) \stackrel{\mathrm{d}}{\Rightarrow} (\bar{m}^t, U) \tag{21}$$

$$(\eta_i^t, v_i) \stackrel{\mathrm{d}}{\Rightarrow} (\bar{\eta}^t, V). \tag{22}$$

The following simple lemma shows the instability of the density evolution recursion.

**Lemma 8.** *Under the density evolution mapping, we obtain the random variables* $(\bar{\eta}', \bar{m}') = \mathsf{DE}(\bar{\eta}, \bar{m}'$ *Let* $\mathsf{m}$ *and* $\mathsf{m}'$ *denote the vector of the first two moments of* $(\bar{\eta}, V, \bar{m}, U)$ *and* $(\bar{\eta}', V', \bar{m}', U')$ *defined as follows:*

$$\mathsf{m} = \left(\mathbb{E}\{V\bar{\eta}\}, \mathbb{E}\{U\bar{m}\}, \mathbb{E}\{\bar{\eta}^2\}, \mathbb{E}\{\bar{m}^2\}\right), \tag{23}$$

*and similarly for* $\mathsf{m}'$. *Then, for* $\|\mathsf{m}\|_2 \to 0$, *we have*

$$\mathsf{m}' = \begin{bmatrix} \lambda^2 & \mu/\gamma & 0 & 0 \\ \mu & 0 & 0 & 0 \\ 0 & 0 & \lambda^2 & \mu/\gamma \\ 0 & 0 & \mu & 0 \end{bmatrix} \mathsf{m} + O(\|\mathsf{m}\|^2) \tag{24}$$

*In particular, the linearized map* $\mathsf{m} \mapsto \mathsf{m}'$ *at* $\mathsf{m} = 0$ *has spectral radius larger than one if and only if* $\lambda^2 + \mu^2/\gamma > 1$.

The interpretation of the cavity prediction and the instability lemma is as follows. If we choose an initialization $(\bar{\eta}^0, V)$, $(\bar{m}^0, U)$ with $\bar{\eta}^0, \bar{m}^0$ *positively correlated* with $V$ and $U$, then this correlation increases exponentially over time if and only if $\lambda^2 + \mu^2/\gamma > 1$[5]. In other words, a small initial correlation is amplified.

While we do not have an initialization that is positively correlated with the true labels, a random initialization $\eta^0, m^0$ has a random correlation with $v, u$ of order $1/\sqrt{n}$. If $\lambda^2 + \mu^2/\gamma > 1$, this correlation is amplified over iterations, yielding a nontrivial reconstruction of $v$. On the other hand, if $\lambda^2 + \mu^2/\gamma < 1$ then this correlation is expected to remain small, indicating that the algorithm does not yield a useful estimate.

## 5 Proof overview

As mentioned above, a key step of our analysis is provided by Theorem 6, which establishes a weak recovery threshold for the Gaussian observation model of Eqs. (8), (9).

The proof proceeds in two steps: first, we prove that, for $\lambda^2 + \mu^2/\gamma < 1$ it is impossible to distinguish between data $A, B$ generated according to this model, and data generated according to the null model $\mu = \lambda = 0$. Denoting by $\mathbb{P}_{\lambda,\mu}$ the law of data $A, B$, this is proved via a standard second moment argument. Namely, we bound the chi square distance uniformly in $n, p$

$$\chi^2(\mathbb{P}_{\lambda,\mu}, \mathbb{P}_{0,0}) \equiv \mathbb{E}_{0,0}\left\{\left(\frac{\mathrm{d}\mathbb{P}_{\lambda,\mu}}{\mathrm{d}\mathbb{P}_{0,0}}\right)^2\right\} - 1 \le C, \tag{25}$$

and then bound the total variation distance by the chi-squared distance $\|\mathbb{P}_{\lambda,\mu} - \mathbb{P}_{0,0}\|_{TV} \le 1 - (\chi^2(\mathbb{P}_{\lambda,\mu}, \mathbb{P}_{0,0}) + 1)^{-1}$. This in turn implies that no test can distinguish between the two hypotheses with probability approaching one as $n, p \to \infty$. The chi-squared bound also allows to show that weak recovery is impossible in the same regime.

In order to prove that weak recovery is possible for $\lambda^2 + \mu^2/\gamma > 1$, we consider the following optimization problem over $x \in \mathbb{R}^n$, $y \in \mathbb{R}^p$:

$$\text{maximize} \quad \langle x, Ax \rangle + b_* \langle x, By \rangle, \tag{26}$$

$$\text{subject to} \quad \|x\|_2 = \|y\|_2 = 1. \tag{27}$$

where $b_* = \frac{2\mu}{\lambda\gamma}$. Denoting solution of this problem by $(\hat{x}, \hat{y})$, we output the (soft) label estimates $\hat{v} = \sqrt{n}\hat{x}$. This definition turns out to be equivalent to the spectral algorithm in the statement of Theorem 6, and is therefore efficiently computable.

This optimization problem undergoes a phase transition exactly at the weak recovery threshold $\lambda^2 + \mu^2/\gamma = 1$, as stated below.

**Lemma 9.** *Denote by $T = T_{n,p}(A, B)$ the value of the optimization problem (26).*

*(i) If $\lambda^2 + \frac{\mu^2}{\gamma} < 1$, then, almost surely*

$$\lim_{n,p \to \infty} T_{n,p}(A, B) = 2\sqrt{1 + \frac{b_*^2\gamma}{4}} + b_*. \tag{28}$$

*(ii) If $\lambda, \mu > 0$, and $\lambda^2 + \frac{\mu^2}{\gamma} > 1$ then there exists $\delta = \delta(\lambda, \mu) > 0$ such that, almost surely*

$$\lim_{n,p \to \infty} T_{n,p}(A, B) = 2\sqrt{1 + \frac{b_*^2\gamma}{4}} + b_* + \delta(\lambda, \mu). \tag{29}$$

*(iii) Further, define*

$$\tilde{T}_{n,p}(\tilde{\delta}; A, B) = \sup_{\|x\|=\|y\|=1, |\langle x, v \rangle| < \tilde{\delta}\sqrt{n}} \left[ \langle x, Ax \rangle + b_* \langle x, By \rangle \right].$$

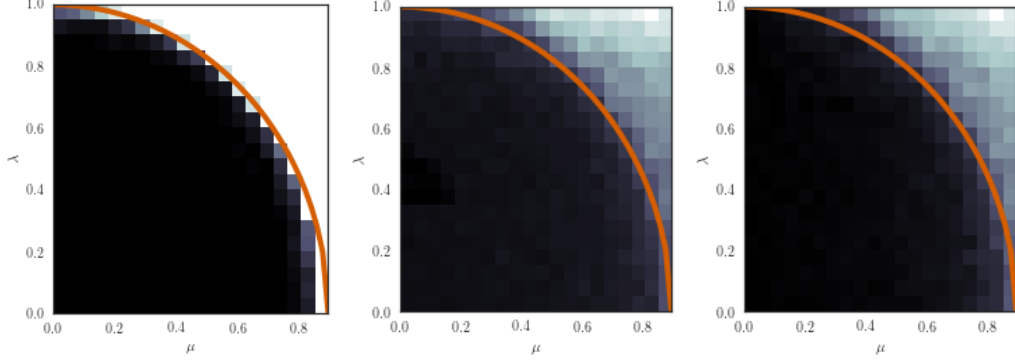

Figure 1: (Left) Empirical probability of rejecting the null (lighter is higher) using BP test. (Middle) Mean overlap $|\langle \widehat{v}^{\mathrm{BP}}, v \rangle / n|$ and (Right) mean covariate overlap $|\langle \widehat{u}^{\mathrm{BP}}, u \rangle|$ attained by BP estimate.

*Then for each $\delta > 0$, there exists $\tilde{\delta} > 0$ sufficiently small, such that, almost surely*

$$\lim_{n,p\to\infty} \tilde{T}_{n,p}(\tilde{\delta}; A, B) < 2\sqrt{1 + \frac{b_*^2 \gamma}{4}} + b_* + \frac{\delta}{2}. \tag{30}$$

The first two points imply that $T_{n,p}(A, B)$ provide a statistic to distinguish between $\mathbb{P}_{0,0}$ and $\mathbb{P}_{\lambda,\mu}$ with probability of error that vanishes as $n, p \to \infty$ if $\lambda^2 + \mu^2/\gamma > 1$. The third point (in conjunction with the second one) guarantees that the maximizer $\widehat{x}$ is positively correlated with $v$, and hence implies weak recovery.

In fact, we prove a stronger result that provides an asymptotic expression for the value $T_{n,p}(A, B)$ for all $\lambda, \mu$. We obtain the above phase-transition result by specializing the resulting formula in the two regimes $\lambda^2 + \mu^2/\gamma < 1$ and $\lambda^2 + \mu^2/\gamma > 1$. We prove this asymptotic formula by Gaussian process comparison, using Sudakov-Fernique inequality. Namely, we compare the Gaussian process appearing in the optimization problem of Eq. (26) with the following ones:

$$\mathcal{X}_1(x, y) = \frac{\lambda}{n}\langle x, v_0 \rangle^2 + \langle x, \widetilde{g}_x \rangle + b_* \sqrt{\frac{\mu}{n}} \langle x, v_0 \rangle \langle y, u_0 \rangle + \langle y, \widetilde{g}_y \rangle, \tag{31}$$

$$\mathcal{X}_2(x, y) = \frac{\lambda}{n}\langle x, v_0 \rangle^2 + \frac{1}{2}\langle x, \widetilde{W}_x x \rangle + b_* \sqrt{\frac{\mu}{n}} \langle x, v_0 \rangle \langle y, u_0 \rangle + \frac{1}{2}\langle y, \widetilde{W}_y y \rangle, \tag{32}$$

where $\widetilde{g}_x, \widetilde{g}_y$ are isotropic Gaussian vectors, with suitably chosen variances, and $\widetilde{W}_x, \widetilde{W}_y$ are GOE matrices, again with properly chosen variances. We prove that $\max_{x,y} \mathcal{X}_1(x, y)$ yields an upper bound on $T_{n,p}(A, B)$, and $\max_{x,y} \mathcal{X}_2(x, y)$ yields a lower bound on the same quantity.

Note that maximizing the first process $\mathcal{X}_1(x, y)$ essentially reduces to solving a separable problem over the coordinates of $x$ and $y$ and hence to an explicit expression. On the other hand, maximizing the second process leads (after decoupling the term $\langle x, v_0 \rangle \langle y, u_0 \rangle$) to two separate problems, one for the vector $x$, and the other for $y$. Each of the two problems reduce to finding the maximum eigenvector of a rank-one deformation of a GOE matrix, a problem for which we can leverage on significant amount of information from random matrix theory. The resulting upper and lower bound coincide asymptotically.

As is often the case with Gaussian comparison arguments, the proof is remarkably compact, and somewhat surprising (it is unclear a priori that the two bounds should coincide asymptotically). While upper bounds by processes of the type of $\mathcal{X}_1(x, y)$ are quite common in random matrix theory, we think that the lower bound by $\mathcal{X}_2(x, y)$ (which is crucial for proving our main theorem) is novel and might have interesting generalizations.

## 6 Experiments

We demonstrate the efficacy of the full belief propagation algorithm, restated below:

$$\eta_i^{t+1} = \sqrt{\frac{\mu}{\gamma}} \sum_{q \in [p]} B_{qi} m_q^t - \frac{\mu}{\gamma} \left( \sum_{q \in [p]} \frac{B_{qi}^2}{\tau_q^t} \right) \tanh(\eta_i^{t-1}) + \sum_{k \in \partial i} f(\eta_{k \to i}^t; \rho) - \sum_{k \in [n]} f(\eta_k^t; \rho_n), \tag{33}$$

$$\eta_{i \to j}^{t+1} = \sqrt{\frac{\mu}{\gamma}} \sum_{q \in [p]} B_{qi} m_q^t - \frac{\mu}{\gamma} \left( \sum_{q \in [p]} \frac{B_{qi}^2}{\tau_q^t} \right) \tanh(\eta_i^{t-1}) + \sum_{k \in \partial i \backslash j} f(\eta_{k \to i}^t; \rho) - \sum_{k \in [n]} f(\eta_k^t; \rho_n), \tag{34}$$

$$m_q^{t+1} = \frac{\sqrt{\mu/\gamma}}{\tau_q^{t+1}} \sum_{j \in [n]} B_{qj} \tanh(\eta_j^t) - \frac{\mu}{\gamma \tau_q^{t+1}} \left( \sum_{j \in [n]} B_{qj}^2 \operatorname{sech}^2(\eta_j^t) \right) m_q^{t-1} \tag{35}$$

$$\tau_q^{t+1} = \left( 1 + \mu - \frac{\mu}{\gamma} \sum_{j \in [n]} B_{qj}^2 \operatorname{sech}^2(\eta_j^t) \right)^{-1}. \tag{36}$$

Here the function $f(;\rho)$ and the parameters $\rho, \rho_n$ are defined as:

$$f(z; \rho) \equiv \frac{1}{2} \log \left( \frac{\cosh(z+\rho)}{\cosh(z-\rho)} \right), \tag{37}$$

$$\rho \equiv \tanh^{-1}(\lambda/\sqrt{d}), \tag{38}$$

$$\rho_n \equiv \tanh^{-1} \left( \frac{\lambda \sqrt{d}}{n-d} \right). \tag{39}$$

We refer the reader to Appendix D for a derivation of the algorithm. As demonstrated in Appendix D, the BP algorithm in Section 4 is obtained by linearizing the above in $\eta$.

In our experiments, we perform 100 Monte Carlo runs of the following process:

1. Sample $A^G, B$ from $\mathbb{P}_{\lambda, \mu}$ with $n = 800, p = 1000, d = 5$.

2. Run BP algorithm for $T = 50$ iterations with random initialization $\eta_i^0, \eta_i^{-1}, m_a^0, m_a^{-1} \sim_{\text{iid}}$ $\mathsf{N}(0, 0.01)$. yielding vertex and covariate iterates $\eta^T \in \mathbb{R}^n, m^T \in \mathbb{R}^p$.

3. Reject the null hypothesis if $\left\| \eta^T \right\|_2 > \left\| \eta^0 \right\|_2$, else accept the null.

4. Return estimates $\widehat{v}_i^{\mathsf{BP}} = \operatorname{sgn}(\eta_i^T)$, $\widehat{u}_a^{\mathsf{BP}} = m_a^T / \left\| m^T \right\|_2$.

Figure 1 (left) shows empirical probabilities of rejecting the null for $(\lambda, \mu) \in [0, 1] \times [0, \sqrt{\gamma}]$. The next two plots display the mean overlap $|\langle \widehat{v}^{\mathsf{BP}}, v \rangle / n|$ and $\langle \widehat{u}^{\mathsf{BP}}, u \rangle / \|u\|$ achieved by the BP estimates (lighter is higher overlap). Below the theoretical curve (red) of $\lambda^2 + \mu^2/\gamma = 1$, the null hypothesis is accepted and the estimates show negligible correlation with the truth. These results are in excellent agreement with our theory. Importantly, while our rigorous result holds only in the limit of diverging $d$, the simulations show agreement already for $d = 5$. This lends further credence to the cavity prediction Claim 3.

## Acknowledgements

A.M. was partially supported by grants NSF DMS-1613091, NSF CCF-1714305 and NSF IIS-1741162. E.M was partially supported by grants NSF DMS-1737944 and ONR N00014-17-1-2598. Y.D would like to acknowledge Nilesh Tripuraneni for discussions about this paper.

## Footnotes

[5]Notice that both the messages variance $\mathbb{E}(\eta^2)$ and covariance with the ground truth $\mathbb{E}(\eta V)$ increase, but the normalized correlation (correlation divided by standard deviation) increases.

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
