[Supplementary Material]

# A   Proof of Theorem 6

We establish Theorem 6 in this section. First, we introduce the notion of contiguity of measures

**Definition 2.** *Let $\{P_n\}$ and $\{Q_n\}$ be two sequences of probability measures on the measurable space $(\Omega_n, \mathcal{F}_n)$. We say that $P_n$ is contiguous to $Q_n$ if for any sequence of events $A_n$ with $Q_n(A_n) \to 0$, $P_n(A_n) \to 0$.*

It is standard that for two sequences of probability measures $P_n$ and $Q_n$ with $P_n$ contiguous to $Q_n$, $\limsup_{n\to\infty} d_{\mathrm{TV}}(P_n, Q_n) < 1$. The following lemma provides sufficient conditions for establishing contiguity of two sequence of probability measures.

**Lemma 10** (see e.g. [MRZ15] ). *Let $P_n$ and $Q_n$ be two sequences of probability measures on $(\Omega_n, \mathcal{F}_n)$. Then $P_n$ is contiguous to $Q_n$ if*

$$\mathbb{E}_{Q_n}\left[\left(\frac{\mathrm{d}P_n}{\mathrm{d}Q_n}\right)^2\right]$$

*exists and remains bounded as $n \to \infty$.*

Our next result establishes that asymptotically error-free detection is impossible below the conjectured detection boundary.

**Lemma 11.** *Let $\lambda, \mu > 0$ with $\lambda^2 + \frac{\mu^2}{\gamma} < 1$. Then $\mathbb{P}_{\lambda,\mu}$ is contiguous to $\mathbb{P}_{0,0}$.*

To establish that consistent detection is possible above this boundary, we need the following lemma. Recall the matrices $A, B$ from the Gaussian model (8), (9).

**Lemma 12.** *Let $b_* = \frac{2\mu}{\lambda\gamma}$. Define*

$$T = \sup_{\|x\|=\|y\|=1} \left[\langle x, Ax \rangle + b_*\langle x, By \rangle\right].$$

*(i) Under $\mathbb{P}_{0,0}$, as $n, p \to \infty$, $T \to 2\sqrt{1 + \frac{b_*^2\gamma}{4}} + b_*$ almost surely.*

*(ii) Let $\lambda, \mu > 0$, $\varepsilon > 0$, with $\lambda^2 + \frac{\mu^2}{\gamma} > 1 + \varepsilon$. Then as $n, p \to \infty$,*

$$\mathbb{P}_{\lambda,\mu}\left(T > 2\sqrt{1 + \frac{b_*^2\gamma}{4}} + b_* + \delta\right) \to 1,$$

*where $\delta := \delta(\varepsilon) > 0$.*

*(iii) Further, define*

$$\tilde{T}(\tilde{\delta}) = \sup_{\|x\|=\|y\|=1, 0<\langle x,v\rangle<\tilde{\delta}\sqrt{n}} \left[\langle x, Ax \rangle + b_*\langle x, By \rangle\right].$$

*Then for each $\delta > 0$, there exists $\tilde{\delta} > 0$ sufficiently small, such that as $n, p \to \infty$,*

$$\mathbb{P}_{\lambda,\mu}\left(\tilde{T}(\tilde{\delta}) < 2\sqrt{1 + \frac{b_*^2\gamma}{4}} + b_* + \frac{\delta}{2}\right) \to 1.$$

We defer the proofs of Lemma 11 and Lemma 12 to Sections A.1 and Section A.5 respectively, and complete the proof of Theorem 6, armed with these results.

*Proof of Theorem 6.* The proof is comparatively straightforward, once we have Lemma 11 and 12. Note that Lemma 11 immediately implies that $\mathbb{P}_{\lambda,\mu}$ is contiguous to $\mathbb{P}_{0,0}$ for $\lambda^2 + \frac{\lambda^2}{\gamma} < 1$.

Next, let $\lambda, \mu > 0$ such that $\lambda^2 + \frac{\mu^2}{\gamma} > 1 + \varepsilon$ for some $\varepsilon > 0$. In this case, consider the test which rejects the null hypothesis $\mathrm{H}_0$ if $T > 2\sqrt{1 + \frac{b_*^2\gamma}{4}} + b_* + \delta$. Lemma 12 immediately implies that the Type I and II errors of this test vanish in this setting.

Finally, we prove that weak recovery is possible whenever $\lambda^2 + \frac{\mu^2}{\gamma} > 1$. To this end, let $(\hat{x}, \hat{y})$ be the maximizer of $\langle x, Ax \rangle + b_*\langle y, Bx \rangle$, with $\|x\| = \|y\| = 1$. Combining parts $(ii)$ and $(iii)$ of Lemma 12, we conclude that $\hat{x}$ achieves weak recovery of the community assignment vector. $\square$

## A.1 Proof of Lemma 11

Fix $\lambda, \mu > 0$ satisfying $\lambda^2 + \frac{\mu^2}{\gamma} < 1$. We start with the likelihood,

$$L(u,v) = \frac{\mathrm{d}\mathbb{P}_{\lambda,\mu}}{\mathrm{d}\mathbb{P}_{0,0}} = L_1(u,v)L_2(u,v),$$

$$L_1(u,v) = \exp\Big[\frac{\lambda}{2}\langle A, vv^T\rangle - \frac{\lambda^2 n}{4}\Big]. \tag{40}$$

$$L_2(u,v) = \exp\Big[p\sqrt{\frac{\mu}{n}}\langle B, uv^T\rangle - \frac{\mu p}{2}\|u\|^2\Big]. \tag{41}$$

We denote the prior joint distribution of (u,v) as $\pi$, and set

$$L_\pi = \mathbb{E}_{(u,v)\sim\pi}\Big[L(u,v)\Big].$$

To establish contiguity, we bound the second moment of $L_\pi$ under the null hypothesis, and appeal to Lemma 10. In particular, we denote $\mathbb{E}_0[\cdot]$ to be the expectation operator under the distribution $P_{(0,0)}$ and compute

$$\mathbb{E}_0[L_\pi^2] = \mathbb{E}_0[\mathbb{E}_{(u_1,v_1),(u_2,v_2)}\big[L(u_1,v_1)L(u_2,v_2)\big]] = \mathbb{E}_{(u_1,v_1),(u_2,v_2)}\Big[\mathbb{E}_0\big[L(u_1,v_1)L(u_2,v_2)\big]\Big],$$

where $(u_1, v_1), (u_2, v_2)$ are i.i.d. draws from the prior $\pi$, and the last equality follows by Fubini's theorem. We have, using (40) and (41),

$$L(u_1,v_1)L(u_2,v_2)$$
$$= \exp\Big[-\frac{\lambda^2 n}{2} - \frac{\mu p}{2n}\big(\|u_1\|^2 + \|u_2\|^2\big) + \frac{\lambda}{2}\big\langle A, v_1 v_1^T + v_2 v_2^T\big\rangle + p\sqrt{\frac{\mu}{n}}\big\langle B, u_1 v_1^T + u_2 v_2^T\big\rangle\Big].$$

Taking expectation under $\mathbb{E}_0[\cdot]$, upon simplification, we obtain,

$$\mathbb{E}_0[L_\pi^2] = \mathbb{E}_{(u_1,v_1),(u_2,v_2)}\Big[\exp\Big[\frac{\lambda^2}{2n}\langle v_1,v_2\rangle^2 + \frac{\mu p}{n}\langle u_1,u_2\rangle\langle v_1,v_2\rangle\Big]\Big] \tag{42}$$

$$= \mathbb{E}_{(u_1,v_1),(u_2,v_2)}\Big[\exp\Big[n\Big(\frac{\lambda^2}{2}\Big(\frac{\langle v_1,v_2\rangle}{n}\Big)^2 + \frac{\mu}{\gamma}\langle u_1,u_2\rangle\frac{\langle v_1,v_2\rangle}{n}\Big)\Big]\Big] \tag{43}$$

$$= \mathbb{E}\Big[\exp\Big[n\Big(\frac{\lambda^2}{2}X^2 + \frac{\mu}{\gamma}XY\Big)\Big]\Big] \tag{44}$$

Here that $X, Y \in [-1,+1]$ are independent, with $X$ distributed as the normalized sum of $n$ Radamacher random variables, and $Y$ as the first coordinate of a uniform vector on the unit sphere. In particular, defining $h(s) = -((1+s)/2)\log((1+s)) - ((1-s)/2)\log((1-s))$, and denoting by $f_Y$ the density of $Y$, we have, for $s \in (2/n)\mathbb{Z}$

$$\mathbb{P}(X = s) = \frac{1}{2^n}\binom{n}{n(1+s/2)} \tag{45}$$

$$\leq \frac{C}{n^{1/2}}e^{nh(s)} \tag{46}$$

$$f_Y(y) = \frac{\Gamma(p/2)}{\Gamma((p-1)/2)\Gamma(1/2)}(1-y^2)^{(p-3)/2} \tag{47}$$

$$\leq C\sqrt{n}(1-y^2)^{p/2}. \tag{48}$$

Approximating sums by integrals, and using $h(s) \leq -s^2/2$, we get

$$\mathbb{E}_0[L_\pi^2] \leq Cn\int_{[-1,1]^2}\exp\Big\{n\Big[\frac{\lambda^2}{2}s^2 + \frac{\mu}{\gamma}sy + h(s) + \frac{1}{2\gamma}\log(1-y^2)\Big]\Big\}\mathrm{d}s\mathrm{d}y \tag{49}$$

$$\leq Cn\int_{\mathbb{R}^2}\exp\Big\{n\Big[\frac{\lambda^2}{2}s^2 + \frac{\mu}{\gamma}sy - \frac{s^2}{2} - \frac{y^2}{2\gamma}\Big]\Big\}\mathrm{d}s\mathrm{d}y \leq C'. \tag{50}$$

The last step holds for $\lambda^2 + \mu^2/\gamma < 1$.

Next, we turn to the proof of Lemma 12. This is the main technical contribution of this paper, and uses a novel Gaussian process comparison argument based on Sudakov-Fernique comparison.

## A.2 A Gaussian process comparison result

Let $Z \sim \mathbb{R}^{p \times n}$ and $W \sim \mathbb{R}^{n \times n}$ denote random matrices with independent entries as follows.

$$W_{ij} \sim \begin{cases} \mathsf{N}(0, \rho/n) & \text{if } i < j \\ \mathsf{N}(0, 2\rho/n) & \text{if } i = j \end{cases} \tag{51}$$

$$\text{where } W_{ij} = W_{ji},$$
$$Z_{ai} \sim \mathsf{N}(0, \tau/p). \tag{52}$$

For an integer $N > 0$, we let $\mathbb{S}^N$ denote the sphere of radius $\sqrt{N}$ in $N$ dimensions, i.e. $\mathbb{S}^N = \{x \in \mathbb{R}^N : \|x\|_2^2 = N\}$. Furthermore let $u_0 \in \mathbb{S}^p$ and $v_0 \in \{\pm 1\}^n$ be fixed vectors. We denote the standard inner product between vectors $x, y \in \mathbb{R}^N$ as $\langle x, y \rangle = \sum_i x_i y_i$. The normalized version will be useful as well: we define $\langle x, y \rangle_N \equiv \sum_i x_i y_i / N$.

We are interested in characterizing the behavior of the following optimization problem in the limit high-dimensional limit $p, n \to \infty$ with constant aspect ratio $n/p = \gamma \in (0, \infty)$.

$$\mathsf{OPT}(\lambda, \mu, b) \equiv \frac{1}{n} \mathbb{E} \max_{(x,y) \in \mathbb{S}^n \times \mathbb{S}^p} \left[ \left( \frac{\lambda}{n} \langle x, v_0 \rangle^2 + \langle x, Wx \rangle \right) + b \left( \sqrt{\frac{\mu}{np}} \langle x, v_0 \rangle \langle y, u_0 \rangle + \langle y, Zx \rangle \right) \right].$$

We now introduce two different comparison processes which give upper and lower bounds to $\mathsf{OPT}(\lambda, \mu, b)$. Their asymptotic values will coincide in the high dimensional limit $n, p \to \infty$ with $n/p = \gamma$. Let $g_x, g_y, W_x$ and $W_y$ be:

$$g_x \sim \mathsf{N}(0, (4\rho + b^2\tau)\mathrm{I}_n) \tag{53}$$

$$g_y \sim \mathsf{N}(0, b^2\tau n/p \mathrm{I}_p), \tag{54}$$

$$(W_x)_{ij} \sim \begin{cases} \mathsf{N}(0, (4\rho + b^2\tau)/n) & \text{if } i < j \\ \mathsf{N}(0, 2(4\rho + b^2\tau)/n) & \text{if } i = j \end{cases} \tag{55}$$

$$(W_y)_{ij} \sim \begin{cases} \mathsf{N}(0, b^2\tau n/p^2) & \text{if } i < j \\ \mathsf{N}(0, 2b^2\tau n/p^2) & \text{if } i = j \end{cases} \tag{56}$$

**Proposition 13.** *We have*

$$\mathsf{OPT}(\lambda, \mu, b) \le \frac{1}{n} \mathbb{E} \max_{(x,y) \in \mathbb{S}^n \times \mathbb{S}^p} \frac{\lambda}{n} \langle x, v_0 \rangle^2 + \langle x, g_x \rangle + b \sqrt{\frac{\mu}{np}} \langle x, v_0 \rangle \langle y, u_0 \rangle + \langle y, g_y \rangle$$

$$\mathsf{OPT}(\lambda, \mu, b) \ge \frac{1}{n} \mathbb{E} \max_{(x,y) \in \mathbb{S}^n \times \mathbb{S}^p} \frac{\lambda}{n} \langle x, v_0 \rangle^2 + \frac{1}{2} \langle x, W_x x \rangle + b \sqrt{\frac{\mu}{np}} \langle x, v_0 \rangle \langle y, u_0 \rangle + \frac{1}{2} \langle y, W_y y \rangle$$
$$\tag{57}$$

*Proof.* The proof is via Sudakov-Fernique inequality. First we compute the distances induced by the three processes. For any pair $(x, y), (x', y')$:

$$\frac{1}{4n} \left( \mathbb{E}\{ (\langle x, Wx \rangle + b\langle y, Zx \rangle - \langle x', Wx' \rangle - b\langle y', Zx' \rangle)^2 \} \right) = \rho(1 - \langle x, x' \rangle_n^2) + \frac{b^2\tau}{2}(1 - \langle x, x' \rangle_n \langle y, y' \rangle_p)$$

$$\frac{1}{n} \left( \mathbb{E}\{ (\langle x, g_x \rangle + \langle y, g_y \rangle - \langle x', g_x \rangle - \langle y', g_y \rangle)^2 \} \right) = 2(4\rho + b^2\tau)(1 - \langle x, x' \rangle_n) + 2b^2\tau(1 - \langle y, y' \rangle_p)$$

$$\frac{1}{4n} \left( \mathbb{E}\{ (\langle x, W_x x \rangle + \langle y, W_y y \rangle - \langle x', W_x x' \rangle - \langle y', W_y y' \rangle)^2 \} \right) = (\rho + \frac{b^2\tau}{4})(1 - \langle x, x' \rangle_n^2) + \frac{b^2\tau}{4}(1 - \langle y, y' \rangle_p^2).$$

This immediately gives:

$$\frac{1}{n}\big(\mathbb{E}\{(\langle x, Wx\rangle + b\langle y, Zx\rangle - \langle x', Wx'\rangle - b\langle y', Zx'\rangle)^2\}\big)-$$
$$\frac{1}{n}\big(\mathbb{E}\{(\langle x, g_x\rangle + \langle y, g_y\rangle - \langle x', g_x\rangle - \langle y, g'_y\rangle)^2\}\big)$$
$$= -4\rho(1 - \langle x, x'\rangle_n)^2 - 2b^2\tau(1 - \langle x, x'\rangle_n)(1 - \langle y, y'\rangle_p) \le 0,$$
$$\frac{1}{4n}\big(\mathbb{E}\{(\langle x, Wx\rangle + b\langle y, Zx\rangle - \langle x', Wx'\rangle - b\langle y', Zx'\rangle)^2\}\big)-$$
$$\frac{1}{4n}\big(\mathbb{E}\{(\langle x, W_x x\rangle + \langle y, W_y y\rangle - \langle x', W_x x\rangle - \langle y', W_y y'\rangle)^2\}\big)$$
$$= \frac{b^2\tau}{4}(\langle x, x'\rangle_n - \langle y, y'\rangle_p)^2 \ge 0.$$

The claim follows. $\qquad\square$

An immediate corollary of this is the following tight characterization for the null value, i.e. the case when $\mu = \lambda = 0$:

**Corollary 14.** *For any $\rho, \tau$ as $n, p$ diverge with $n/p \to \gamma$, we have*

$$\lim_{n\to\infty} \mathsf{OPT}(0,0) = \sqrt{4\rho + b^2\tau} + b\sqrt{\frac{\tau}{\gamma}} \tag{58}$$

Note that this upper bound generalizes the maximum eigenvalue and singular value bounds of $W$, $Z$ respectively. In particular, the case $\tau = 0$ corresponds to the maximum eigenvalue of $W$, which yields $\mathsf{OPT} = 2\sqrt{\rho}$ while the maximum singular value of $Z$ can be recovered by setting $\rho$ to 0 and $b$ to 1, yielding $\mathsf{OPT} = \sqrt{\tau}(1 + \gamma^{-1/2})$. Corollary 14 demonstrates the limit for the case when $\mu = \lambda = 0$. The following theorem gives the limiting value when $\lambda, \mu$ may be nonzero.

**Theorem 15.** *Suppose $\mathsf{G} : \mathbb{R} \times \mathbb{R}_+ \to \mathbb{R}$ is as follows:*

$$\mathsf{G}(\kappa, \sigma^2) = \begin{cases} \kappa/2 + \sigma^2/2\kappa & \text{if } \kappa^2 \ge \sigma^2, \\ \sigma & \text{otherwise.} \end{cases} \tag{59}$$

*Then the optimal value $\mathsf{OPT}(\lambda, \mu)$ is*

$$\lim_{n\to\infty} \mathsf{OPT}(\lambda, \mu) = \min_{t \ge 0} \left\{ \mathsf{G}(2\lambda + b\mu t, 4\rho + b^2\tau) + \gamma^{-1}\mathsf{G}(b/t, b^2\gamma\tau) \right\}. \tag{60}$$

*If the minimum above occurs at $t = t_*$ such that $\mathsf{G}'(2\lambda + b\mu t_*, 4\rho + b^2\tau) = \partial_\kappa \mathsf{G}(\kappa, 4\rho + b^2\tau)|_{\kappa=2\lambda+b\mu t_*} > 0$, then $\lim_{n\to\infty} \mathsf{OPT}(\lambda, \mu) > \sqrt{4\rho + b^2\tau} + \gamma^{-1}\sqrt{\frac{\tau}{\gamma}}$.*

### A.3 Proof of Theorem 15: the upper bound

The following lemma removes the effect of the projection of $g_x$ ($g_y$) along $v_0$ (resp. $u_0$). Let $F(x, y) = \frac{1}{n}[\lambda x_1^2 + \langle x, g_x\rangle + b\sqrt{\mu}x_1 y_1 + \langle y, g_y\rangle]$. Further, let $\widetilde{g}_x$ ($\widetilde{g}_y$) be the vectors obtained by setting the first coordinate of $g_x$ (resp. $g_y$) to zero, and $\widetilde{F}(x, y) = \frac{1}{n}[\lambda x_1^2 + \langle x, \widetilde{g}_x\rangle + b\sqrt{\mu}x_1 y_1 + \langle y, \widetilde{g}_y\rangle]$.

**Lemma 16.** *The optima of $F$ and $\widetilde{F}$ differ by at most $o(1)$. More precisely:*

$$\left| \mathbb{E} \max_{x,y} F(x, y) - \mathbb{E} \max_{x,y} \widetilde{F}(x, y) \right| = O\Big(\frac{1}{\sqrt{n}}\Big).$$

*Proof.* For any $x, y$:

$$F(x, y) = \frac{1}{n}\Big(\lambda x_1^2 + \langle x, g_x\rangle + \sqrt{\mu}x_1 y_1 + \langle y, g_y\rangle\Big) = \widetilde{F}(x, y) + \frac{1}{n}(x_1(g_x)_1 + y_1(g_y)_1)$$
$$\left| F(x, y) - \widetilde{F}(x, y) \right| \le \frac{1}{n}(\sqrt{n}|(g_x)_1| + \sqrt{p}|(g_y)_1|).$$

Maximizing each side over $x, y$ and taking expectation yields the lemma. $\qquad\square$

With this in hand, we can concentrate on computing the maximum of $\widetilde{F}(x, y)$.

**Lemma 17.** *Let $\widetilde{g}_x$ ($\widetilde{g}_y$) be the projection of $g_x$ (resp. $g_y$) orthogonal to the first basis vector. Then*

$$\limsup_{n\infty} \mathbb{E} \max_{(x,y)\in\mathbb{S}^n\times\mathbb{S}^p} \widetilde{F}(x,y) \leq \min_{t\leq 0} \mathsf{G}(2\lambda + b\mu t, 4\rho + b^2\tau) + \frac{1}{\gamma}\mathsf{G}(b/t, b^2\gamma\tau) \qquad (61)$$

*Proof.* Since $\widetilde{F}(x, y)$ increases if we align the signs of $x_1$ and $y_1$ to $+1$, we can assume that they are positive. Furthermore, for fixed, positive $x_1, y_1$, $\widetilde{F}$ is maximized if the other coordinates align with $\widetilde{g}_x$ and $\widetilde{g}_y$ respectively. Therefore:

$$\max_{x,y} \widetilde{F}(x,y) = \max_{x_1\in[0,\sqrt{n}],y_1\in[0,\sqrt{p}]} \frac{\lambda x_1^2}{n} + \sqrt{1 - \frac{x_1^2}{n}}\frac{\|\widetilde{g}_x\|}{\sqrt{n}} + \frac{b\sqrt{\mu x_1 y_1}}{n} + \sqrt{1 - \frac{y_1^2}{p}}\frac{\sqrt{p}\,\|\widetilde{g}_y\|}{n}$$

$$= \max_{m_1,m_2\in[0,1]} \lambda m_1 + \sqrt{1 - m_1}\frac{\|\widetilde{g}_x\|}{\sqrt{n}} + b\sqrt{\frac{\mu m_1 m_2 p}{n}} + \sqrt{1 - m_2}\frac{\sqrt{p}\,\|\widetilde{g}_y\|}{n}$$

$$\leq \max_{m_1,m_2\in[0,1]} \left(\lambda + \frac{b\mu t}{2}\right)m_1 + \sqrt{1 - m_1}\frac{\|\widetilde{g}_x\|}{\sqrt{n}} + \frac{p}{n}\left(\frac{bm_2}{2t} + \sqrt{1 - m_2}\frac{\|\widetilde{g}_y\|}{\sqrt{p}}\right)$$

$$= \mathsf{G}(2\lambda + b\mu t, \|\widetilde{g}_x\|^2/n) + \frac{1}{\gamma}\mathsf{G}\left(\frac{b}{t}, \|\widetilde{g}_y\|^2/p\right), \qquad (62)$$

where the first equality is change of variables, the second inequality is the fact that $2\sqrt{ab} = \min_{t\geq 0}(at + b/t)$, and the final equality is by direct calculus.

Now let $t_*$ be any minimizer of $\mathsf{G}(2\lambda + b\mu t, 4\rho + b^2\tau) + \gamma^{-1}\mathsf{G}(b/t, b^2\gamma\tau)$. We may assume that $t_* \notin \{0, \infty\}$, otherwise we can use $t_*(\varepsilon)$, an $\varepsilon$-approximate minimizer in $(0, \infty)$ in the argument below. Since the above holds for any $t$, we have:

$$\max_{x,y} \widetilde{F}(x,y) \leq \mathsf{G}(2\lambda + b\mu t_*, \|\widetilde{g}_x\|^2/n) + \gamma^{-1}\mathsf{G}(b/t_*, \|\widetilde{g}_y\|^2/p). \qquad (63)$$

By the strong law of large numbers, $\|\widetilde{g}_x\|^2/n \to 4\rho + b^2\tau$ and $\|\widetilde{g}_y\|^2/p \to b^2\gamma\tau$ almost surely. Further, as $\mathsf{G}(\kappa, \sigma^2)$ is continuous in the second argument on $(0, \infty)$, when $\kappa \notin \{0, \infty\}$, almost surely:

$$\limsup_{x,y} \max \widetilde{F}(x,y) \leq \mathsf{G}(2\lambda + b\mu t_*, 4\rho + b^2\tau) + \gamma^{-1}\mathsf{G}(b/t_*, b^2\gamma\tau). \qquad (64)$$

Taking expectations and using bounded convergence yields the lemma. □

We can now prove the upper bound.

*Theorem 15, upper bound.* Using Proposition 13, Lemma 16 and Lemma 17 in order:

$$\mathsf{OPT}(\lambda, \mu) \leq \mathbb{E}\{\max_{x,y} F(x,y)\} \qquad (65)$$

$$\leq \mathbb{E}\{\max_{x,y} \widetilde{F}(x,y)\} + o(n^{-1/3}) \qquad (66)$$

$$\leq \min_t \mathsf{G}(2\lambda + b\mu t, 4\rho + b^2\tau) + \frac{1}{\gamma}\mathsf{G}(b/t, b^2\gamma\tau) + o(n^{-1/3}). \qquad (67)$$

Taking limit $p \to \infty$ yields the result. □

### A.4 Proof of Theorem 15: the lower bound

Recall that $t_*$ denotes the optimizer of the upper bound $\mathsf{G}(2\lambda + b\mu t, 4\rho + b^2\tau) + \gamma^{-1}\mathsf{G}(b/t, b^2\gamma\tau)$. By stationarity, we have:

$$b\mu\mathsf{G}'(2\lambda + b\mu t_*, 4\rho + b^2\tau) - \frac{b}{\gamma t_*^2}\mathsf{G}'(\frac{b}{t_*}, b^2\gamma\tau) = 0. \qquad (68)$$

Now we proceed in two cases. First, suppose $\mathsf{G}'(2\lambda + b\mu t_*, 4\rho + b^2\tau) = 0$. In this case $\mathsf{G}'(b/t_*, b^2\gamma\tau)/t_*^2 = 0$, whence $\mathsf{G}'(b/t_*, b^2\gamma\tau) = 0$. Indeed, the case when $t_* = \infty$ also satisfies this. However, this also implies that $2\lambda + b\mu t_* \leq \sqrt{4\rho + b^2\tau}$ and $t_* \geq (\gamma\tau)^{-1/2}$, whereby $\mathsf{G}(2\lambda + b\mu t_*, 4\rho + b^2\tau) = \sqrt{4\rho + b^2\tau}$ and $\mathsf{G}'(b/t_*, b^2\gamma\tau) = b\sqrt{\gamma\tau}$. In this case we consider $\tilde{x}, \tilde{y}$ to be the principal eigenvectors of $W_x, W_y$ rescaled to norms $\sqrt{n}, \sqrt{p}$ respectively and, hence using (57),

$$\mathsf{OPT}(\lambda, \mu, b) \geq \frac{1}{2n}\mathbb{E}\Big[\langle \tilde{x}, W_x\tilde{x}\rangle + \langle \tilde{y}, W_y\tilde{y}\rangle\Big] - o(1). \tag{69}$$

By standard results on GOE matrices the right hand side converges to $\sqrt{4\rho + b^2\tau} + b\sqrt{\frac{\tau}{\gamma}}$ implying the required lower bound.

Now consider the case that $\mathsf{G}'(2\lambda + b\mu t_*, 4\rho + b^2\tau) > 0$. Importantly, by stationarity we have

$$t_*^2 = \frac{\mathsf{G}'(bt_*^{-1}, b^2\gamma\tau)}{\mu\gamma\mathsf{G}'(2\lambda + b\mu t_*, 4\rho + b^2\tau)}, \tag{70}$$

and that $t_*$ is finite since the numerator is decreasing in $t_*$. The key ingredient to prove the lower bound is the following result on the principal eigenvalue/eigenvector of a deformed GOE matrix.

**Theorem 18** ([CDMF+09, KY13]). *Suppose $W \in \mathbb{R}^{n\times n}$ is a GOE matrix with variance $\sigma^2$, i.e. $W_{ij} = W_{ji} \sim \mathsf{N}(0, (1+\delta_{ij}\sigma^2/p)$ and $A = \kappa v_0 v_0^{\mathsf{T}} + W$ where $v_0$ is a unit vector. Then the following holds almost surely and in expectation:*

$$\lim_{n\to\infty} \lambda_1(A) = 2\mathsf{G}(\kappa, \sigma^2) = \begin{cases} 2\sigma & \text{if } \kappa < \sigma \\ \kappa + \sigma^2/\kappa & \text{if } \kappa > \sigma. \end{cases} \tag{71}$$

$$\lim_{n\to\infty} \langle v_1(A), v_0\rangle^2 = 2\mathsf{G}'(\kappa, \sigma^2) = \begin{cases} 0 & \text{if } \kappa < \sigma, \\ 1 - \sigma^2/\kappa^2 & \text{if } \kappa > \sigma. \end{cases} \tag{72}$$

*where $\mathsf{G}'$ denotes the derivative with respect to the first argument.*

For the prescribed $t_*$, define:

$$H(x, y) = \Big(\lambda + \frac{b\mu t_*}{2}\Big)\frac{\langle x, v_0\rangle^2}{n^2} + \frac{\langle x, W_x x\rangle}{2n} + \frac{p}{n}\Big(\frac{b\langle y, u_0\rangle^2}{2t_* p^2} + \frac{\langle y, W_y y\rangle}{2p}\Big) \tag{73}$$

Let $\tilde{x}, \tilde{y}$ be the principal eigenvector of $(2\lambda + b\mu t_*)v_0 v_0^{\mathsf{T}}/n + W_x, bt_*^{-1}u_0 u_0^{\mathsf{T}}/p + W_y$, rescaled to norm $\sqrt{n}$ and $\sqrt{p}$ respectively. Further, we choose the sign of $\tilde{x}$ so that $\langle \tilde{x}, v_0\rangle \geq 0$, and analogously for $\tilde{y}$. Now, fixing an $\varepsilon > 0$, we have by Theorem 18, for every $p$ large enough:

$$H(\tilde{x}, \tilde{y}) \geq \mathsf{G}(2\lambda + b\mu t_*, 4\rho + b^2\tau) + \gamma^{-1}\mathsf{G}(bt_*^{-1}, b^2\gamma\tau) - \varepsilon \tag{74}$$

$$\frac{\langle \tilde{x}, v_0\rangle}{n} = \sqrt{2\mathsf{G}'(2\lambda + b\mu t_*, 4\rho + b^2\tau)} + O(\varepsilon) \tag{75}$$

$$\frac{\langle \tilde{y}, u_0\rangle}{p} = \sqrt{2\mathsf{G}'(bt_*^{-1}, b^2\gamma\tau)} + O(\varepsilon) \tag{76}$$

We have, therefore:

$$\mathsf{OPT}(\lambda, \mu, b) \geq \mathbb{E}\Big[H(\tilde{x}, \tilde{y}) + \Big(\frac{b}{n}\sqrt{\frac{\mu}{np}}\langle \tilde{x}, v_0\rangle\langle \tilde{y}, u_0\rangle - \frac{b\mu t\langle \tilde{x}, v_0\rangle^2}{2n^2} - \frac{b\langle y, u_0\rangle^2}{2tnp}\Big)\Big] \tag{77}$$

$$\geq \mathsf{G}(2\lambda + b\mu t_*, 4\rho + b^2\tau) + \gamma^{-1}\mathsf{G}(bt_*^{-1}, b^2\gamma\tau) + O(\varepsilon(t_* \vee t_*^{-1}))$$

$$+ \Big(2\sqrt{\frac{\mu}{\gamma}\mathsf{G}'(2\lambda + b\mu t_*, 4\rho + b^2\tau)\mathsf{G}'(bt_*^{-1}, b^2\gamma\tau)} - b\mu t_*\mathsf{G}'(2\lambda + b\mu t_*, 4\rho + b^2\tau)$$

$$- \frac{\mathsf{G}'(bt_*^{-1}, b^2\gamma\tau)}{\gamma t_*}\Big)$$

$$\geq \mathsf{G}(2\lambda + b\mu t_*, 4\rho + b^2\tau) + \gamma^{-1}\mathsf{G}(bt_*^{-1}, b^2\gamma\tau) + O(\varepsilon(t_* \vee t_*^{-1})). \tag{78}$$

Here the first inequality since we used a specific guess $\tilde{x}, \tilde{y}$, the second using Theorem 18 and the final inequality follows since the remainder term vanishes due to Eq. (70). Taking expectations and letting $\varepsilon$ going to 0 yields the required lower bound.

Given Corollary 14 and Theorem 15, it is not too hard to establish Lemma 12, which we proceed to do next.

## A.5 Proof of Lemma 12

Recall $b_* = \frac{2\mu}{\lambda\gamma}$. Part (i) follows directly from Corollary 14, upon setting $\rho = \tau = 1$, and $b = b_*\sqrt{\gamma}$. To establish part (ii), we use Theorem 15. In particular, it suffices to establish that with this specific choice of $b = b_*\sqrt{\gamma}$, for any $(\lambda, \mu)$ with $\lambda^2 + \mu^2/\gamma > 1$, the minimizer $t_*$ of $G(2\lambda + b\mu t, 4 + b^2) + \gamma^{-1}G(b/t, b^2\gamma)$ satisfies $G'(2\lambda + b\mu t_*, 4 + b^2) > 0$. Let us assume, if possible, that $G(2\lambda + b\mu t_*, 4 + b^2) = 0$. Using the stationary point condition (68), in this case $G'(b/t_*, b^2\gamma) = 0$. Next, using the definition of $G$ (59), observe that this implies

$$t_* > \frac{1}{\sqrt{\gamma}}, \quad 2\lambda + \frac{2\mu^2}{\lambda\sqrt{\gamma}}t_* < \sqrt{4 + \frac{4\mu^2}{\lambda^2\gamma}}.$$

These imply:

$$\frac{2}{\lambda}\left(\lambda^2 + \frac{\mu^2}{\gamma}\right) < 2\lambda + 2\frac{\mu^2 t_*}{\lambda\mu\sqrt{\gamma}} \tag{79}$$

$$< \sqrt{4 + \frac{4\mu^2}{\lambda^2\gamma}} \tag{80}$$

$$= \frac{2}{\lambda}\sqrt{\lambda^2 + \frac{\mu^2}{\gamma}}. \tag{81}$$

That this is impossible whenever $\lambda^2 + \frac{\mu^2}{\gamma} > 1$. This establishes part (ii). To establish part (iii), we again use the upper bound from Proposition 13, and note that for $0 < \langle x, v \rangle < \tilde{\delta}\sqrt{n}$,

$$\mathbb{E}[\tilde{T}(\tilde{\delta})] \leq \lambda\tilde{\delta}^2 + \sqrt{4 + b_*^2} + \max_{\|y\|=1}\{b_*\sqrt{\mu}\tilde{\delta}\langle u, y \rangle + \frac{1}{\gamma}\langle y, g \rangle\},$$

where $g \sim \mathsf{N}(0, b^2\gamma I_p/p)$. The proof follows using continuity in $\tilde{\delta}$. This completes the proof.

## B  Proof of Lemma 8

Recall the distributional recursion specified by density evolution (Definition 1).

$$\bar{m}'|_U \stackrel{d}{=} \mu U \mathbb{E}[V\bar{\eta}] + \zeta_1\sqrt{\mu\mathbb{E}[\bar{\eta}^2]},$$

$$\bar{\eta}'|_{V'=+1} \stackrel{d}{=} \frac{\lambda}{\sqrt{d}}\left[\sum_{k=1}^{k_+}\bar{\eta}_k|_+ + \sum_{k=1}^{k_-}\bar{\eta}_k|_-\right] - \lambda\sqrt{d}\mathbb{E}[\bar{\eta}] + \frac{\mu}{\gamma}\mathbb{E}[U\bar{m}] + \zeta_2\sqrt{\frac{\mu}{\gamma}\mathbb{E}[\bar{m}^2]},$$

where $V \sim U(\{\pm 1\})$, $U \sim \mathsf{N}(0,1)$, $k_+ \sim \text{Poisson}\left(\frac{d+\lambda\sqrt{d}}{2}\right)$, $k_- \sim \text{Poisson}\left(\frac{d-\lambda\sqrt{d}}{2}\right)$, $\zeta_1, \zeta_2 \sim \mathsf{N}(0,1)$ are all mutually independent. Further, $\{\bar{\eta}_k|_+\}$ are iid random variables, distributed as $\bar{\eta}|_{V=+1}$. Similarly, $\{\bar{\eta}_k|_-\}$, are iid random variables, distributed as $\bar{\eta}|_{V=-1}$. Finally, we require the collections to be mutually independent, and independent of the other auxiliary variables defined above.

Given these distributional recursions, we compute the vector of moments

$$\mathbb{E}[V'\bar{\eta}'] = \lambda^2\mathbb{E}[V\bar{\eta}] + \frac{\mu}{\gamma}\mathbb{E}[U\bar{m}]$$

$$\mathbb{E}[U'\bar{m}'] = \mu\mathbb{E}[V\bar{\eta}]$$

$$\mathbb{E}[\bar{\eta}'^2] = \lambda^2\mathbb{E}[\bar{\eta}^2] + \frac{\mu^2}{\gamma^2}\mathbb{E}^2[U\bar{m}] + \frac{\mu}{\gamma}\mathbb{E}[\bar{m}^2] + 2\frac{\lambda^2}{\gamma}\mathbb{E}[U\bar{m}]\mathbb{E}[V\bar{\eta}].$$

$$\mathbb{E}[\bar{m}'^2] = \mu^2\mathbb{E}^2[V\bar{\eta}] + \mu\mathbb{E}[\bar{\eta}^2]$$

Thus the induced mapping on moments $\phi^{\mathsf{DE}} : \mathbb{R}^4 \to \mathbb{R}^4$, $\phi^{\mathsf{DE}}(z_1, z_2, z_3, z_4) = (\phi_1, \phi_2, \phi_3, \phi_4)$, with

$$\phi_1 = \lambda^2 z_1 + \frac{\mu}{\gamma} z_2$$

$$\phi_2 = \mu z_1$$

$$\phi_3 = \frac{\mu^2}{\gamma^2} z_2^2 + \frac{2\lambda^2}{\gamma} z_1 z_2 + \lambda^2 z_3 + \frac{\mu}{\gamma} z_4,$$

$$\phi_4 = \mu^2 z_1^2 + \mu z_3.$$

The Jacobian of $\phi^{\mathsf{DE}}$ at 0 is, up to identical row/column permutation:

$$J = \begin{bmatrix} \lambda^2 I_2 & \frac{\mu}{\gamma} I_2 \\ \mu I_2 & 0 \end{bmatrix}.$$

By direct computation, we see that $z$ is an eigenvalue of $J$ if and only if $z^2 - \lambda^2 z - \frac{\mu^2}{\gamma} = 0$. Consider the quadratic function $f(z) = z^2 - \lambda^2 z - \frac{\mu^2}{\gamma}$ and note that $f(0) < 0$. Thus to check whether $f$ has a root with magnitude greater than 1, it suffices to check its value at $z = 1, -1$. Note that if $\lambda^2 + \frac{\mu^2}{\gamma} > 1$, $f(1) < 0$ and thus $J$ has an eigenvalue greater than 1. Conversely, if $\lambda^2 + \frac{\mu^2}{\gamma} < 1$, $f(1) > 0$ and $f(-1) = 1 + \lambda^2 - \frac{\mu^2}{\gamma} > 1 - \frac{\mu^2}{\gamma} > 0$. This completes the proof.

## C  Proof of Theorem 4

We prove Theorem 4 in this Section. Recall the matrix mean square errors

$$\mathsf{MMSE}(v; A, B) = \frac{1}{n(n-1)} \mathbb{E}\Big[ \|vv^T - \mathbb{E}[vv^T | A, B]\|_F^2 \Big],$$

$$\mathsf{MMSE}(v; A^G, B) = \frac{1}{n(n-1)} \mathbb{E}\Big[ \|vv^T - \mathbb{E}[vv^T | A^G, B]\|_F^2 \Big].$$

The following lemma is immediate from Lemma 4.6 in [DAM16].

**Lemma 19.** *Let $\widehat{v} = \widehat{v}(A, B)$ be any estimator so that $\|\widehat{v}\|_2 = \sqrt{n}$. Then*

$$\liminf_{n \to \infty} \frac{\langle \widehat{v}, v \rangle}{n} > 0 \text{ in probability} \Rightarrow \limsup_{n \to \infty} \mathsf{MMSE}(v; A, B) < 1. \tag{82}$$

*Furthermore, if $\limsup_{n \to \infty} \mathsf{MMSE}(v; A, B) < 1$, there exists an estimator $\widehat{s}(A, B)$ with $\|\widehat{s}(A, B)\|_2 = \sqrt{n}$ so that, in probability:*

$$\liminf_{n \to \infty} \frac{\langle \widehat{s}, v \rangle}{n} > 0. \tag{83}$$

*Indeed, the same holds for the observation model $A^G, B$.*

*Proof of Theorem 4.* Consider first the case $\lambda^2 + \frac{\mu^2}{\gamma} < 1$. For any $\theta \in [0, \lambda]$, $\theta^2 + \mu^2/\gamma < 1$ as well. Suppose we have $A(\theta), B$ according to model (8), (9) where $\lambda$ is replaced with $\theta$. By Theorem 6 (applied at $\theta$) and the second part of Lemma 19, $\liminf_{n \to \infty} \mathsf{MMSE}(v; A(\theta), B) = 1$. Using the I-MMSE identity [GSV05], this implies

$$\lim_{n \to \infty} \frac{1}{n} (I(v; A(\theta), B) - I(v; A(0), B)) = \frac{\theta^2}{4}. \tag{84}$$

By Theorem 5, for all $\theta \in [0, \lambda]$

$$\lim_{d \to \infty} \lim_{n \to \infty} \frac{1}{n} (I(v; A^G(\theta), B) - I(v; A^G(0), B)) = \frac{\theta^2}{4}, \tag{85}$$

$$\text{and, therefore } \lim_{n \to \infty} \mathsf{MMSE}(v; A^G, B) = 1 \tag{86}$$

This implies, via the first part of Lemma 19 that for any estimator $\widehat{v}(A^G; B)$, we have $\limsup_{n \to \infty} |\langle \widehat{v}, v \rangle| / n = 0$ in probability, as required.

Conversely, consider the case $\lambda^2 + \frac{\mu^2}{\gamma} > 1$. We may assume that $\mu^2/\gamma < 1$, as otherwise the result follows from Theorem 2. Let $\lambda_0 = (1 - \mu^2/\gamma)^{1/2}$.

Now, by the same argument for Eqs.(84), (85), we obtain for all $\theta_1, \theta_2 \in [\lambda_0, \lambda]$:

$$\limsup_{n \to \infty} \frac{1}{n}(I(v; A(\theta_1), B) - I(v; A(\theta_2), B)) < \frac{\theta_1^2 - \theta_2^2}{4}. \tag{87}$$

Applying Theorem 5, we have for all $\theta_1, \theta_2, \theta \in [\lambda_0, \lambda]$:

$$\lim_{d \to \infty} \limsup_{n \to \infty} \frac{1}{n}(I(v; A^G(\theta_1), B) - I(v; A^G(\theta_2), B)) < \frac{\theta_1^2 - \theta_2^2}{4} \tag{88}$$

$$\text{and therefore, } \limsup \mathsf{MMSE}(v; A^G(\theta), B) < 1. \tag{89}$$

Applying then Lemma 19 implies that we have an estimator $\widehat{s}(A^G, B)$ with non-trivial overlap i.e. in probability:

$$\lim_{d \to \infty} \liminf_{n \to \infty} \frac{\langle \widehat{s}, v \rangle}{n} > 0. \tag{90}$$

This completes the proof.

$\square$

# D   Belief propagation: derivation

In this section we will derive the belief propagation algorithm. Recall the observation model for $(A^G, B) \in \mathbb{R}^{n \times n} \times \mathbb{R}^{p \times n}$ in Eqs. (1), (2):

$$A_{ij}^G = \begin{cases} 1 & \text{with probability } \frac{d + \lambda\sqrt{d}v_iv_j}{n} \\ 0 & \text{otherwise.} \end{cases} \tag{91}$$

$$B_{qi} = \sqrt{\frac{\mu}{n}}u_q v_i + Z_{qi}, \tag{92}$$

where $u_q$ and $Z_{qi}$ are independent $\mathsf{N}(0, 1/p)$ variables.

We will use the following conventions throughout this section to simplify some of the notation. We will index nodes in the graph, i.e. elements in $[n]$ with $i, j, k \ldots$ and covariates, i.e. elements in $[p]$ with $q, r, s, \ldots$. We will use '$\simeq$' to denote equality of probability distributions (or densities) up to an omitted proportionality constant, that may change from line to line. We will omit the superscript $G$ in $A^G$. In the graph $G$, we will denote neighbors of a node $i$ with $\partial i$ and non-neighbors with $\partial i^c$.

We start with the posterior distribution of $u, v$ given the data $A, B$:

$$d\mathbb{P}\{u, v | A, B\} = \frac{d\mathbb{P}\{A, B | u, v\}}{d\mathbb{P}\{A, B\}} d\mathbb{P}\{u, v\} \tag{93}$$

$$\simeq \prod_{i<j} \left(\frac{d + \lambda\sqrt{d}v_iv_j}{n}\right)^{A_{ij}} \left(1 - \frac{d + \lambda\sqrt{d}v_iv_j}{n}\right)^{1-A_{ij}}$$

$$\cdot \prod_{q,i} \exp\left(\sqrt{\frac{\mu p^2}{n}} B_{qi} u_q v_i\right) \prod_q \exp\left(-\frac{p(1+\mu)}{2} u_q^2\right). \tag{94}$$

The belief propagation algorithm operates 'messages' $\nu_{i \to j}^t, \nu_{q \to i}^t, \nu_{i \to q}^t$ which are probability distributions. They represent the marginals of the variables $v_i, u_q$ in the absence of variables $v_j, u_q$, in the posterior distribtuion $d\mathbb{P}\{u, v | A, B\}$. We denote by $\mathbb{E}_{i \to j}^t, \mathbb{E}_{q \to i}^t, \mathbb{E}_{i \to q}^t$ expectations with respect to

these distributions. The messages are are computed using the following update equations:

$$\nu_{i\to j}^{t+1}(v_i) \simeq \prod_{q\in[p]} \mathbb{E}_{q\to i}^t \Big\{ \exp\Big(\sqrt{\frac{\mu p^2}{n}} B_{qi} v_i u_q\Big)\Big\} \prod_{k\in\partial i\setminus j} \mathbb{E}_{k\to i}^t\Big(\frac{d+\lambda\sqrt{d}v_i v_k}{n}\Big) \prod_{k\in\partial i^c\setminus j} \mathbb{E}_{k\to i}^t\Big(1 - \frac{d+\lambda\sqrt{d}v_i v_k}{n}\Big),$$

(95)

$$\nu_{i\to q}^{t+1}(v_i) \simeq \prod_{r\in[p]\setminus q} \mathbb{E}_{r\to i}^t \Big\{ \exp\Big(\sqrt{\frac{\mu p^2}{n}} B_{ri} v_i u_r\Big)\Big\} \prod_{k\in\partial i} \mathbb{E}_{k\to i}^t\Big(\frac{d+\lambda\sqrt{d}v_i v_k}{n}\Big) \prod_{k\in\partial i^c} \mathbb{E}_{k\to i}^t\Big(1 - \frac{d+\lambda\sqrt{d}v_i v_k}{n}\Big),$$

(96)

$$\nu_{q\to i}^{t+1}(u_q) \simeq \exp\Big(-\frac{p(1+\mu)u_q^2}{2}\Big) \prod_{j\neq i} \mathbb{E}_{j\to q}^t\Big\{ \exp\Big(\sqrt{\frac{\mu p^2}{n}} B_{qj} v_j u_q\Big)\Big\}.$$

(97)

As is standard, we define $\nu_i^t, \nu_q^t$ in the same fashion as above, except without the removal of the incoming message.

## D.1 Reduction using Gaussian ansatz

The update rules (95), (96), (97) are in terms of probability distributions, i.e. measures on the real line or $\{\pm 1\}$. We reduce them to update rules on real numbers using the following analytical ansatz. The measure $\nu_{i\to j}^t$ on $\{\pm 1\}$ can be summarized using the log-odds ratio:

$$\eta_{i\to j}^t \equiv \frac{1}{2} \log \frac{\nu_{i\to j}^t(+1)}{\nu_{i\to j}^t(-1)},$$

(98)

and we similarly define $\eta_{i\to q}^t, \eta_i^t$. In order to reduce the densities $\nu_{q\to i}^t$, we use the Gaussian ansatz:

$$\nu_{q\to i}^t = \mathsf{N}\Big(\frac{m_{q\to i}^t}{\sqrt{p}}, \frac{\tau_{q\to i}^t}{p}\Big).$$

(99)

With Equations (98) and (99) we can now simplify Equations (95) to (97). The following lemma computes the inner marginalizations in Equations (95) to (97). We omit the proof.

**Lemma 20.** *With $\nu^t, \mathbb{E}^t$ as defined as per Equations* (95) *to* (97) *and $\eta^t, m^t, \tau^t$ as in Equations* (98) *and* (99) *we have*

$$\mathbb{E}_{q\to i}^t \exp\Big(\sqrt{\frac{\mu p^2}{n}} B_{qi} v_i u_q\Big) = \exp\Big(\sqrt{\frac{\mu p}{n}} B_{qi} v_i m_{q\to i}^t + \frac{\mu p}{2n} B_{qi}^2 \tau_{q\to i}^t\Big),$$

(100)

$$\mathbb{E}_{i\to j}^t\Big(\frac{d+\lambda\sqrt{d}v_i v_j}{n}\Big) = \frac{d}{n}\Big(1 + \frac{\lambda v_j}{\sqrt{d}} \tanh(\eta_{i\to j}^t)\Big),$$

(101)

$$\mathbb{E}_{i\to j}^t\Big(1 - \frac{d+\lambda\sqrt{d}v_i v_j}{n}\Big) = 1 - \frac{d}{n}\Big(1 + \frac{\lambda v_j}{\sqrt{d}} \tanh(\eta_{i\to j}^t)\Big),$$

(102)

$$\mathbb{E}_{i\to q}^t \exp\Big(p\sqrt{\frac{\mu}{n}} B_{qi} v_i u_q\Big) = \frac{\cosh(\eta_{i\to q}^t + p\sqrt{\mu/n} B_{qi} u_q)}{\cosh \eta_{i\to q}^t}.$$

(103)

The update equations take a simple form using the following definitions

$$f(z;\rho) \equiv \frac{1}{2} \log\Big(\frac{\cosh(z+\rho)}{\cosh(z-\rho)}\Big),$$

(104)

$$\rho \equiv \tanh^{-1}(\lambda/\sqrt{d}),$$

(105)

$$\rho_n \equiv \tanh^{-1}\Big(\frac{\lambda\sqrt{d}}{n-d}\Big).$$

(106)

With this, we first compute the update equation for the node messages $\eta^{t+1}$. Using Equations (95), (96) and (100) to (103):

$$\eta_{i\to j}^{t+1} = \sqrt{\frac{\mu}{\gamma}} \sum_{q\in[p]} B_{qi} m_{q\to i}^t + \sum_{k\in\partial i\setminus j} f(\eta_{k\to i}^t;\rho) - \sum_{k\in\partial i\setminus j} f(\eta_{k\to i}^t;\rho_n), \tag{107}$$

$$\eta_{i\to q}^{t+1} = \sqrt{\frac{\mu}{\gamma}} \sum_{r\in[p]\setminus q} B_{ri} m_{r\to i}^t + \sum_{k\in\partial i} f(\eta_{k\to i}^t;\rho) - \sum_{k\in\partial i^c} f(\eta_{k\to i}^t;\rho_n), \tag{108}$$

$$\eta_i^{t+1} = \sqrt{\frac{\mu}{\gamma}} \sum_{q\in[p]} B_{qi} m_{q\to i}^t + \sum_{k\in\partial i} f(\eta_{k\to i}^t;\rho) - \sum_{k\in\partial i^c} f(\eta_{k\to i}^t;\rho_n). \tag{109}$$

Now we compute the updates for $m_{a\to i}^t, \tau_{a\to i}^t$. We start from Equations (97) and (100), and use Taylor approximation assuming $u_q, B_{jq}$ are both $O(1/\sqrt{p})$, as the ansatz (99) suggests.

$$\log \nu_{q\to i}^{t+1}(u_q) = \text{const.} + \frac{-p(1+\mu)}{2} u_q^2 + \sum_{j\in[n]\setminus i} \log\cosh\left(\eta_{j\to q}^t + p\sqrt{\frac{\mu}{n}} B_{qj} u_q\right) \tag{110}$$

$$= \text{const.} + \frac{-p(1+\mu)}{2} u_q^2 + \left(p\sqrt{\frac{\mu}{n}} \sum_{j\in[n]\setminus i} B_{qj}\tanh(\eta_{j\to q}^t)\right) u_q + \left(\frac{p^2\mu}{2n} \sum_{j\in[n]} B_{qj}^2 \text{sech}^2(\eta_{j\to q}^t)\right) u_q^2 + O\left(\frac{1}{\sqrt{n}}\right). \tag{111}$$

Note that here we compute $\log\nu^{t+1}$ only up to constant factors (with slight abuse of the notation '$\simeq$'). It follows from this quadratic approximation that:

$$\tau_{q\to i}^{t+1} = \left(1 + \mu - \frac{\mu}{\gamma} \sum_{j\in[n]\setminus i} B_{qj}^2 \text{sech}^2(\eta_{j\to q}^t)\right)^{-1}, \tag{112}$$

$$m_{q\to i}^{t+1} = \tau_{q\to i}^{t+1} \sqrt{\frac{\mu}{\gamma}} \sum_{j\in[n]\setminus i} B_{qj}\tanh(\eta_{j\to q}^t) \tag{113}$$

$$= \frac{\sqrt{\mu/\gamma} \sum_{j\in[n]\setminus i} B_{qj}\tanh(\eta_{j\to q}^t)}{1 + \mu - \mu\gamma^{-1} \sum_{j\in[n]} B_{qj}^2 \text{sech}^2(\eta_{j\to q}^t)}. \tag{114}$$

Updates computing $m_q^{t+1}, \tau_q^{t+1}$ are analogous.

### D.2 From message passing to approximate message passing

The updates for $\eta^t, m^t$ derived in the previous section require keeping track of $O(np)$ messages. In this section, we further reduce the number of messages to $O(dn + p)$, i.e. linear in the size of the input graph observation.

The first step is to observe that the dependence of $\eta_{i\to j}^t$ on $j$ is negligible when $j$ is not a neighbor of $i$ in the graph $G$. This derivation is similar to the presentation in [DKMZ11]. As $\sup_{z\in\mathbb{R}} f(z;\rho) \le \rho$. Therefore, if $i, j$ are not neighbors in $G$:

$$\eta_{i\to j}^t = \eta_i^t - f(\eta_{j\to i}^{t-1};\rho_n) \tag{115}$$

$$= \eta_i^t + O(\rho_n) = \eta_i^t + O\left(\frac{1}{n}\right). \tag{116}$$

Now, for a pair $i, j$ not connected, by Taylor expansion and the fact that $\partial_z f(z;\rho) \le \tanh(\rho)$,

$$f(\eta_{i\to j}^t;\rho_n) - f(\eta_i^t;\rho_n) = O\left(\frac{\tanh(\rho_n)}{n}\right) = O\left(\frac{1}{n^2}\right). \tag{117}$$

Therefore, the update equation for $\eta_{i\to j}^{t+1}$ satisfies:

$$\eta_{i\to j}^{t+1} = \sqrt{\frac{\mu}{\gamma}} \sum_{q\in[p]} B_{qi} m_{q\to i}^t + \sum_{k\in\partial i\setminus j} f(\eta_{k\to i}^t;\rho) - \sum_{k\in[n]} f(\eta_k^t;\rho_n) + O\left(\frac{1}{n}\right), \tag{118}$$

$$\eta_i^{t+1} = \eta_{i\to j}^{t+1} + f(\eta_{j\to i}^t;\rho). \tag{119}$$

Similarly for $\eta_{i \to q}^{t+1}$ we have:

$$\eta_{i \to q}^{t+1} = \sqrt{\frac{\mu}{\gamma}} \sum_{r \in [p] \setminus q} B_{ri} m_{r \to i}^{t} + \sum_{k \in \partial i} f(\eta_{k \to i}^{t}; \rho) - \sum_{k \in [n]} f(\eta_k^t; \rho_n) + O\left(\frac{1}{n}\right). \quad (120)$$

Ignoring $O(1/n)$ correction term, the update equations reduce to variables $(\eta_{i \to j}^t, \eta_i^t)$ where $i, j$ are neighbors.

We now move to reduce updates for $\eta_{i \to q}^t$ and $m_{q \to i}^t$ to involving $O(n)$ variables. This reduction is more subtle then that of $\eta_{i \to j}^t$, where we are able to simply ignore the dependence of $\eta_{i \to j}^t$ on $j$ if $j \notin \partial i$. We follow a derivation similar to that in [Mon12]. We use the ansatz:

$$\eta_{i \to q}^t = \eta_i^t + \delta \eta_{i \to q}^t \quad (121)$$
$$m_{q \to i}^t = m_q^t + \delta m_{q \to i}^t \quad (122)$$
$$\tau_{q \to i}^t = \tau_q^t + \delta \tau_{q \to i}^t, \quad (123)$$

where the corrections $\delta \eta_{i \to q}^t, \delta m_{q \to i}^t, \delta \tau_{q \to i}^t$ are $O(1/\sqrt{n})$. From Equations (97) and (120) at iteration $t$:

$$\eta_i^t + \delta \eta_{i \to q}^t = \sqrt{\frac{\mu}{\gamma}} \sum_{r \in [p] \setminus q} B_{ri}(m_r^{t-1} + \delta m_{r \to i}^{t-1}) + \sum_{k \in \partial i} f(\eta_{k \to i}^{t-1}; \rho) - \sum_k f(\eta_k^{t-1}; \rho_n) \quad (124)$$

$$= \sqrt{\frac{\mu}{\gamma}} \sum_{r \in [p]} B_{ri}(m_r^{t-1} + \delta m_{r \to i}^{t-1}) + \sum_{k \in \partial i} f(\eta_{k \to i}^{t-1}; \rho) - \sum_k f(\eta_k^{t-1}; \rho_n) - \sqrt{\frac{\mu}{\gamma}}\left(B_{qi} m_q^{t-1} + B_{qi} \delta m_{q \to i}^{t-1}\right). \quad (125)$$

Notice that the last term is the only term that depends on $q$. Further, since $B_{qi} \delta m_{q \to i}^{t-1} = O(1/n)$ by our ansatz, we may safely ignore it to obtain

$$\eta_i^t = \sqrt{\frac{\mu}{\gamma}} \sum_{r \in [p]} B_{ri}(m_r^{t-1} + \delta m_{r \to i}^{t-1}) + \sum_{k \in \partial i} f(\eta_{k \to i}^{t-1}; \rho) - \sum_k f(\eta_k^{t-1}; \rho_n) \quad (126)$$

$$\delta \eta_{i \to q}^t = -\sqrt{\frac{\mu}{\gamma}} B_{qi} m_q^{t-1}. \quad (127)$$

We now use the update equation for $\tau_{q \to i}^{t+1}$:

$$\tau_q^{t+1} = \left(1 + \mu - \frac{\mu}{\gamma} \sum_{j \in [n]} B_{qj}^2 \operatorname{sech}^2(\eta_j^t + \delta \eta_{j \to q}^t)\right)^{-1} + O(1/n) \quad (128)$$

$$= \left(1 + \mu - \frac{\mu}{\gamma} \sum_{j \in [n]} B_{qj}^2 \left((\operatorname{sech}^2(\eta_j^t) - 2\operatorname{sech}^2(\eta_j^t) \tanh(\eta_j^t) \delta \eta_{i \to q}^t)\right)\right)^{-1} + O(1/n), \quad (129)$$

where we expanded the equation to linear order in $\delta \eta_{i \to q}^t$ and ignored higher order terms. By the identification Equation (127):

$$\tau_q^{t+1} = \left(1 + \mu - \frac{\mu}{\gamma} \sum_{j \in [n]} B_{qj}^2 \operatorname{sech}^2(\eta_j^t) + 2\left(\frac{\mu}{\gamma}\right)^{3/2} \sum_{j \in [n]} B_{qj}^3 \operatorname{sech}^2(\eta_j^t) \tanh(\eta_j^t) m_q^{t-1}\right)^{-1} + O(1/n). \quad (130)$$

Notice here, that there is no term that explicitly depends on $i$ and the final term is $O(1/\sqrt{n})$ since $B_{qj} = O(1/\sqrt{n})$. Therefore, ignoring lower order terms, we have the identification:

$$\tau_q^{t+1} = \left(1 + \mu - \frac{\mu}{\gamma} \sum_{j \in [n]} B_{qj}^2 \operatorname{sech}^2(\eta_j^t)\right)^{-1}, \quad (131)$$

$$\delta \tau_{q \to i}^{t+1} = 0. \quad (132)$$

Now we simplify the update for $m_{q \to i}^{t+1}$ using Taylor expansion to first order:

$$m_q^{t+1} + \delta m_{q \to i}^{t+1} = \frac{\sqrt{\mu/\gamma}}{\tau_q^{t+1}} \sum_{j \in [n] \setminus i} B_{qj} \tanh(\eta_j^t + \delta \eta_{j \to q}^t) \tag{133}$$

$$= \frac{\sqrt{\mu/\gamma}}{\tau_q^{t+1}} \sum_{j \in [n] \setminus i} \left( B_{qj} \tanh(\eta_j^t) + B_{qj} \operatorname{sech}^2(\eta_i^t) \delta \eta_{j \to q}^t \right) \tag{134}$$

$$= \frac{\sqrt{\mu/\gamma}}{\tau_q^{t+1}} \sum_{j \in [n] \setminus i} \left( B_{qj} \tanh(\eta_j^t) - \sqrt{\frac{\mu}{\gamma}} B_{qj}^2 \operatorname{sech}^2(\eta_j^t) m_q^{t-1} \right) \tag{135}$$

$$= \frac{\sqrt{\mu/\gamma}}{\tau_q^{t+1}} \sum_{j \in [n]} B_{qj} \tanh(\eta_j^t) - \frac{\mu}{\gamma \tau_q^{t+1}} \left( \sum_{j \in [n]} B_{qj}^2 \operatorname{sech}^2(\eta_j^t) \right) m_q^{t-1}$$

$$- \frac{\sqrt{\mu/\gamma}}{\tau_q^{t+1}} \left( B_{qi} \tanh(\eta_i^t) - \sqrt{\mu/\gamma} B_{qi}^2 \operatorname{sech}^2(\eta_i^t) m_q^{t-1} \right). \tag{136}$$

Only the final term is dependent on $i$, therefore we can identify:

$$m_q^{t+1} = \frac{\sqrt{\mu/\gamma}}{\tau_q^{t+1}} \sum_{j \in [n]} B_{qj} \tanh(\eta_j^t) - \frac{\mu}{\gamma \tau_q^{t+1}} \left( \sum_{j \in [n]} B_{qj}^2 \operatorname{sech}^2(\eta_j^t) \right) m_q^{t-1}, \tag{137}$$

$$\delta m_{q \to i}^{t+1} = -\frac{\sqrt{\mu/\gamma}}{\tau_q^{t+1}} B_{qi} \tanh(\eta_i^t). \tag{138}$$

Here, as before, we ignore the lower order term in $\delta m_{q \to i}^{t+1}$. Now we can substitute the identification Equation (138) back in Equation (126) at iteration $t+1$:

$$\eta_i^{t+1} = \sqrt{\frac{\mu}{\gamma}} \sum_{r \in [p]} B_{ri} m_r^t - \frac{\mu}{\gamma} \sum_{r \in [p]} \frac{B_{ri}^2}{\tau_r^t} \tanh(\eta_i^{t-1}) + \sum_{k \in \partial i} f(\eta_{k \to i}^t; \rho) - \sum_k f(\eta_k^t; \rho_n). \tag{139}$$

Collecting the updates for $\eta_i^t, \eta_{i \to j}^t, m_q^t$ we obtain the approximate message passing algorithm:

$$\eta_i^{t+1} = \sqrt{\frac{\mu}{\gamma}} \sum_{q \in [p]} B_{qi} m_q^t - \frac{\mu}{\gamma} \left( \sum_{q \in [p]} \frac{B_{qi}^2}{\tau_q^t} \right) \tanh(\eta_i^{t-1}) + \sum_{k \in \partial i} f(\eta_{k \to i}^t; \rho) - \sum_{k \in [n]} f(\eta_k^t; \rho_n),$$
$$\tag{140}$$

$$\eta_{i \to j}^{t+1} = \sqrt{\frac{\mu}{\gamma}} \sum_{q \in [p]} B_{qi} m_q^t - \frac{\mu}{\gamma} \left( \sum_{q \in [p]} \frac{B_{qi}^2}{\tau_q^t} \right) \tanh(\eta_i^{t-1}) + \sum_{k \in \partial i \setminus j} f(\eta_{k \to i}^t; \rho) - \sum_{k \in [n]} f(\eta_k^t; \rho_n),$$
$$\tag{141}$$

$$m_q^{t+1} = \frac{\sqrt{\mu/\gamma}}{\tau_q^{t+1}} \sum_{j \in [n]} B_{qj} \tanh(\eta_j^t) - \frac{\mu}{\gamma \tau_q^{t+1}} \left( \sum_{j \in [n]} B_{qj}^2 \operatorname{sech}^2(\eta_j^t) \right) m_q^{t-1} \tag{142}$$

$$\tau_q^{t+1} = \left( 1 + \mu - \frac{\mu}{\gamma} \sum_{j \in [n]} B_{qj}^2 \operatorname{sech}^2(\eta_j^t) \right)^{-1}. \tag{143}$$

## D.3 Linearized approximate message passing

This algorithm results from expanding the updates Equations (140) to (143) to linear order in the messages $\eta_i^t, \eta_{i\to j}^t$:

$$\eta_i^{t+1} = \sqrt{\frac{\mu}{\gamma}} \sum_{q\in[p]} B_{qi} m_q^t - \frac{\mu}{\gamma}\left(\sum_{q\in[p]} \frac{B_{qi}^2}{\tau_q^t}\right)\eta_i^{t-1} + \frac{\lambda}{\sqrt{d}}\sum_{k\in\partial i}\eta_{k\to i}^t - \frac{\lambda\sqrt{d}}{n}\sum_{k\in[n]}\eta_k^t \tag{144}$$

$$\eta_{i\to j}^{t+1} = \sqrt{\frac{\mu}{\gamma}} \sum_{q\in[p]} B_{qi} m_q^t - \frac{\mu}{\gamma}\left(\sum_{q\in[p]} \frac{B_{qi}^2}{\tau_q^t}\right)\eta_i^{t-1} + \frac{\lambda}{\sqrt{d}}\sum_{k\in\partial i\setminus j}\eta_{k\to i}^t - \frac{\lambda\sqrt{d}}{n}\sum_{k\in[n]}\eta_k^t \tag{145}$$

$$m_q^{t+1} = \frac{\sqrt{\mu/\gamma}}{\tau_q^{t+1}} \sum_{j\in[n]} B_{qj}\eta_j^t - \frac{\mu}{\gamma\tau_q^{t+1}}\left(\sum_{j\in[n]} B_{qj}^2\right)m_q^{t-1} \tag{146}$$

$$\tau_q^{t+1} = \left(1 + \mu - \frac{\mu}{\gamma}\sum_{j\in[n]} B_{qj}^2\right)^{-1}. \tag{147}$$

This follows from the linear approximation $f(z;\rho) = \tanh(\rho)z$ for small $z$. The algorithm given in the main text follows by using the law of large numbers to approximate $\sum_{j\in[n]} B_{qj}^2 \approx 1/\gamma$, $\sum_{q\in[p]} B_{qj}^2 \approx 1$, and hence $\tau 4_q \approx 1$.