[Reviews · NeurIPS 2018]

Reviewer 1



This paper considers the problem of community structure recovery (in the balanced binary case) when presented with informative node covariates. The authors propose a simple statistical model for this problem and conjecture a sharp threshold for detecting the latent structure in this model (using the non-rigorous cavity method). They then rigorously establish the validity of this threshold in a particular limiting regime. Finally, they provide an algorithm for solving this problem and demonstrate empirical support for the above claims. The paper is very well written and is bound to be of interest to the NIPS community. I did not check all the proofs in detail, but I performed some sanity checks to verify their validity. I recommend this paper be accepted to the NIPS program. I have one suggestion for the authors: Please include an intuitive discussion about the threshold derived in Claim 3, right after the statement. This will be helpful for the reader.

Reviewer 2



UPDATE after authors' rebuttal I am quite happy about the scientific answers provided by the authors. Hence I upgrade my score from 6 to 7. Hoping it will be enough for an acceptance to NIPS (competition is harsh!), and if not, still hoping to read your work somewhere! This paper is about the clustering of statistical entities for which network data and individual measurements are available. The article reads well and is technically sound. he numerical experiments are disappointing, I don't really understand how they illustrate the performance of the method. I would have expected some accuracy measure for the methods on its ability to identify meaningful clusters. Moreover, I challenge the novelty of the approach presented here. It was easy to find references like https://www.ncbi.nlm.nih.gov/pmc/articles/PMC5890472/ (very similar problem with the same kind of data) or less recently https://ieeexplore.ieee.org/document/4359897/ (with the implementation the authors published in https://www.ncbi.nlm.nih.gov/pmc/articles/PMC3051335/, you could well compare to your approach). I truly feel you should compare the method you present to the state-of-the-art literature. To me, your model resembles a Hidden Conditional Random Fields (or a Markov Random Fields) via Equations (1) and (2). Details: - l17, the A data is not necessarily a 'similarity' measure, it could just be a graph measure directly defined, e.g. regulatory relationships. A similarity measure is more a representation of similar measures on nodes, hence more a proxy for a B data in your settings. - l65: you focus on bi-partitioning, which in graph is clearly an easier (yet challenging) special case of the general framework. Why is that? How can your method be applied with K classes and K>2? - I quite like the 'in Other words...' explanation of our theorems. - What is a 'Claim' for you? A Proposition? A conjecture? - Again, your algorithm (Section 4) could be compared to traditional approaches like Variational Inference and MCMC strategies. - In the reference, make the style uniform: with or without first names?

Reviewer 3



This paper considered the problem of detecting latent community structure given a sparse graph along with high-dimensional node covariates. The problems with either sparse graph or high-dimensional node covariates have been rather extensively studied in the literature, so the modeling innovation is community detection based on both pieces of information correlated with the same latent structure. To me, this problem seems to be rather contrived. Not surprisingly, the threshold prediction is one (see equation (4)) that naturally bridges the two extreme cases where only one piece of information is available (see Theorem 1 and 2). From the technical viewpoint, the main result is a formal confirmation of the threshold prediction in the limit of large degrees via a Gaussian model. The proof is rather nontrivial and appears to be interesting.